# Establishment of a Cell Suspension Culture of *Eysenhardtia platycarpa*: Phytochemical Screening of Extracts and Evaluation of Antifungal Activity

**DOI:** 10.3390/plants10020414

**Published:** 2021-02-23

**Authors:** Antonio Bernabé-Antonio, Alejandro Sánchez-Sánchez, Antonio Romero-Estrada, Juan Carlos Meza-Contreras, José Antonio Silva-Guzmán, Francisco Javier Fuentes-Talavera, Israel Hurtado-Díaz, Laura Alvarez, Francisco Cruz-Sosa

**Affiliations:** 1Departamento de Madera, Celulosa y Papel, Centro Universitario de Ciencias Exactas e Ingenierías, Universidad de Guadalajara, Km 15.5 Carretera Guadalajara-Nogales, Col. Las Agujas, Zapopan 45100, Jalisco, Mexico; alexcucba.bio@gmail.com (A.S.-S.); are@uaem.mx (A.R.-E.); jcmezac@academicos.udg.mx (J.C.M.-C.); jasilva@dmcyp.cucei.udg.mx (J.A.S.-G.); francisco.fuentes@academicos.udg.mx (F.J.F.-T.); ihurtado@uaem.mx (I.H.-D.); 2Centro de Investigaciones Químicas-IICBA, Universidad Autónoma del Estado de Morelos, Av. Universidad No. 1001, Col. Chamilpa, Cuernavaca 62209, Morelos, Mexico; lalvarez@uaem.mx; 3Departamento de Biotecnología, División de Ciencias Biológicas y de la Salud, Universidad Autónoma Metropolitana-Unidad Iztapalapa, San Rafael Atlixco 186, Col. Vicentina, Ciudad de México 09340, Mexico

**Keywords:** plant in vitro culture, plant extracts, gas chromatography, hexadecanoic acid, antifungal activity

## Abstract

*Eysenhardtia platycarpa* (*Fabaceae*) is a medicinal plant used in Mexico. Biotechnological studies of its use are lacking. The objective of this work was to establish a cell suspension culture (CSC) of *E. platycarpa*, determine the phytochemical constituents by spectrophotometric and gas chromatography‒mass spectrometry (GC‒MS) methods, evaluate its antifungal activity, and compare them with the intact plant. Friable callus and CSC were established with 2 mg/L 1-naphthaleneacetic acid plus 0.1 mg/L kinetin. The highest total phenolics of CSC was 15.6 mg gallic acid equivalents (GAE)/g dry weight and the total flavonoids content ranged from 56.2 to 104.1 µg quercetin equivalents (QE)/g dry weight. The GC‒MS analysis showed that the dichloromethane extracts of CSC, sapwood, and heartwood have a high amount of hexadecanoic acid (22.3–35.3%) and steroids (13.5–14.7%). Heartwood and sapwood defatted hexane extracts have the highest amount of stigmasterol (~23.4%) and *β*-sitosterol (~43%), and leaf extracts presented *β*-amyrin (16.3%). Methanolic leaf extracts showed mostly sugars and some polyols, mainly D-pinitol (74.3%). Compared with the intact plant, dichloromethane and fatty hexane extracts of CSC exhibited percentages of inhibition higher for *Sclerotium cepivorum*: 71.5% and 62.0%, respectively. The maximum inhibition for *Rhizoctonia solani* was with fatty hexane extracts of the sapwood (51.4%). Our study suggests that CSC extracts could be used as a possible complementary alternative to synthetic fungicides.

## 1. Introduction

For centuries, plants have been an important source of natural products for humans; they have been used as flavorings and condiments and for treating health disorders and preventing diseases, including epidemics [1]. Plant products have formed the basis for many useful pharmaceuticals and agrochemicals, and through their rational use, plants can be a potential alternative for obtaining extracts or bioactive compounds to control several diseases in both humans and crops [2,3]. However, many of these extracts or compounds are isolated from wild plants, whose collection generally has a negative impact on the environment [4]. Furthermore, agricultural soil is increasingly limited and, in the future, the production of plants in the field will not be feasible. Although plants can be cultivated in the field, many of them need several years to be harvested, and in many cases, the yields of bioactive compounds and biological activity of cultivated or wild plants are lower compared with cultured plant cells [5,6]. Plant cell culture is a biotechnological tool that has the potential to accelerate the production of natural products in a controlled environment; in addition, cell culture provides a renewable source of natural products, since plant cell culture can be produced and harvested at all times of the year [3,4].

In this regard, *Eysenhardtia platycarpa* (*Fabaceae*) is a wild plant extensively exploited as firewood, fodder, or to manufacture utensils and furniture such as “equipales” and fences; in addition, in traditional Mexican medicine, an infusion prepared from the wood is used against kidney and gallbladder diseases [7,8,9]. All these uses, as well as forest fires, are causing a decline in wild populations. Among the few existing investigations on *E. platycarpa*, all of them have used the wild plant as a source of pharmacological studies [10]. For instance, flavonoids with cytotoxic and antibacterial activity were isolated from methanolic extracts of branches and leaves of *E. platycarpa* [11]. The in vivo anti-inflammatory activity of flavones isolated from the leaves has also been reported [12]. However, no effort has been made to carry out studies aimed at the sustainable use of this species. In a biotechnological study developed for *Eysenhardtia polystachya* (a closely related species), the antifungal activity of cell suspension culture extracts was reported against *R. solani* and *S. cepivorum* [13]. To our knowledge, however, there are no reports involving biotechnological studies of *E. platycarpa*. Therefore, it is necessary to look for biotechnological techniques that allow us to obtain bioactive extracts while preserving natural diversity and the environment.

The main aim of this study was to establish a cell suspension culture from *E. platycarpa* internodal segments. We also performed the phytochemical characterization of the CSC and intact plant extracts through the spectrophotometric quantification of total phenolics and flavonoids content and, by GC‒MS analysis, the volatile natural compounds or silylated derivatives were determined. Furthermore, the biological activity of the extracts against the phytopathogenic fungi *Rhizoctonia solani* and *Sclerotium cepivorum* was explored. As a hypothesis, we expected that the *E. platycarpa* extracts produced by cell suspension cultures would present comparable antifungal activity with respect to extracts from the wild plant, as well as the compounds identified by GC‒MS.

## 2. Results and Discussion

### 2.1. Obtaining Plantlets and Callus Induction

*In vitro* cultures of *Eysenhardtia platycarpa* are shown in Figure 1. The seed germination rate was 98% at 10 days of culture and no microbial contamination was observed. The in vitro plants grew easily without plant growth regulators after being transferred to 1-L jars (Figure 1A).

In other *Fabaceae* species such as *Prosopis laevigata*, the in vitro germination can occur after 3–7 days in mechanically scarified seeds [14,15]. Regarding callus induction, a suitable response of callus formation was exhibited on internodal segments after 15 days of culture (Figure 1B). All treatments had percentages of callus induction greater than 50.0%, regardless of the type of auxins (NAA: naphthaleneacetic acid; 2,4-D: 2,4‒dichlorophenoxyacetic acid) or kinetin (KIN) as a cytokinin (Table 1).

The control (PGRs-free) showed callus formation in 50% of segments, but exhibited scarce growth, showed a brown color, and later died. It is possible that *E. platycarpa* contains auxin in leaves and stems, which may explain why, even in the control treatment, the segments formed a callus. In fact, it has been reported that indoleacetic acid (IAA) occurs naturally in plants, mainly in young leaves or seeds [16,17]. Moreover, it is known that levels of naturally occurring auxin in explant tissues depend on the mother plant from which the explants were taken [18].

In internodal segments of *E. platycarpa*, all evaluated treatments showed a positive effect on callus formation; 62.0% of the treatments with NAA and KIN formed a callus on 100% of the segments, mainly with NAA (1.0 or 2.0 mg/L), regardless of KIN concentration. Calluses of this treatment were clear greenish and more friable in appearance than the other treatments; moreover, the calluses had homogeneous growth during all subcultures. The other treatments exhibited a smaller and semicompact callus, and most of them did not show growth. In the case of treatments with 2,4-D plus KIN, it was observed that, as the concentration of 2,4-D increased, there was a trend of an increased percentage of callus induction, and this generally occurred in the presence of KIN (Table 1). Similarly, treatments with 2,4-D (1.0 or 2.0 mg/L) combined with KIN (0.1 or 2.0 mg/L) had 100% of explants with calluses. The remaining treatments, including the control, showed calluses with percentages between 50.0% and 94.0%. Calluses of the best treatments with both auxins, i.e., NAA (2.0 mg/L) with KIN (0.1 mg/L) or 2,4-D (2.0 mg/L) with KIN (0.1 mg/L), were subcultured periodically for six months to increase callus production.

In a study reported for *E. polystachya*, it was found that callus induction was variable, according to the PGRs, i.e., percentages of calluses between 65.6% and 98.4% in leaf explants were obtained with picloram (PIC) plus KIN, and from 64.1% to 100% with NAA plus KIN [13]. In studies carried out on *P. laevigata*, calluses from cotyledons, hypocotyls, and root explants were obtained with 2,4-D plus 6-benzylaminopurine (BAP) or KIN, in percentages of 28.0% to 100% [15]. This indicates that combining auxins and cytokinins plays an indispensable role in inducing and increasing the percentage of calluses [19].

### 2.2. Cell Suspension Cultures

#### 2.2.1. Growth Kinetics and Sucrose Consumption

Calluses induced with NAA (2.0 mg/L) and KIN (0.1 mg/L) or 2,4-D (2.0 mg/L) and KIN (0.1 mg/L) were the best treatments, showing friable characteristics, and were used to initiate the establishment of the cell suspension cultures (CSC) of *E. platycarpa* with the same plant growth regulators; however, during one month of culture, cells cultured in MS liquid medium with 2,4-D (2.0 mg/L) and KIN (0.1 mg/L) showed poor growth. Therefore, this treatment was discarded from the experiment. In contrast, cells cultured with NAA (2.0 mg/L) and KIN (0.1 mg/L) exhibited growth and an abundant accumulation of biomass (Figure 1C–E).

The growth kinetics of *E. platycarpa* was maintained for 18 days, during which it exhibited typical growth (Figure 2). According to the modified Gompertz model, the lag phase [λ (days)] lasted 1.84 days. The exponential phase was six days (from day 2 to day 8). The stationary phase was observed between days 10 and 12; then, the senescence phase was gradually observed between days 14 and 18. In addition, the maximum accumulation of biomass dry weight (18.62 g/L DW) occurred after 10 days and the growth index was 5.35, obtaining a yield of 0.621 g dry biomass/g sucrose. By the natural logarithm of biomass, the specific growth rate (*μ*) was 0.24 days^–1^ and the doubling time (td) was 2.92 days, while, using the modified Gompertz model, *μ*_max_ was 0.25 days^–1^ and X_max_ was 19.6 (g/L) with a correlation coefficient (R) of 0.982. This model has been reported for callus and cell suspension cultures of *Jatropha curcas* [20,21] and plant growth [22].

During the adaptation phase of *E. platycarpa*, the amounts of sugars remained unchanged; however, in the exponential growth phase, there was a greater demand for sugars and, consequently, a higher production of biomass. From day 12, the sugars in the culture medium were almost completely consumed; this coincided with the stationary phase and the senescence phase. Sugars were not detected in the culture medium from day 12; however, it is likely that they are present in undetectable traces since the culture was not significantly affected in the senescence phase. This behavior has also been reported in other species such as *C. brasiliense* cell suspension cultures [23]. This may be due to the cells accumulating glucose and fructose and using them for metabolism [24]. In fact, GC‒MS showed that the methanolic extract of cell suspension cultures had large amounts of disaccharides (more than 40%).

In other species such as *P. laevigata*, the treatment with 2,4-D (1.5 mg/L) and KIN (1.0 mg/L) was the most suitable for establishing the CSC [15]. It is known from callus induction treatments on semisolid culture medium that these do not always adapt best to a liquid culture medium. For instance, *P. laevigata* leaf explants presented 100% of callus with trichlorophenoxyacetic acid (2,4,5-T, 1.28 mg/L) with BAP (1.13 mg/L) or 2,4,5-T (1.28 mg/L) with KIN (1.08 mg/L); however, only calluses containing KIN were suitable for establishing the CSC [14]. This indicates that, even among closely related species, the genotype is an influential factor in the response of cell cultures; moreover, the PGRs activity varies depending on the presence of transporter or receptor biosynthesized proteins in the explants, affecting in vitro culture development [25]. In another study conducted in *E. polystachya*, the maximum accumulation of dry biomass was 14 g/L after 10 days of culture [13]. Biomass yield was lower than that reported in the present study for *E. platycarpa* at the same time. Studies conducted by Maldonado-Magaña et al. [15] in *P. laevigata* also found a dry biomass yield of 15.6 g/L after 21 days of culture.

#### 2.2.2. Total Phenolics and Flavonoids Content

It has been reported that phenolics (TPH), flavonoids (TFL), and other compounds of plant extracts are effective against phytopathogenic fungi [26,27]; therefore, it is desirable to quantify these groups of compounds in the extracts. Because the cell cultures were obtained from leaf explants, in this part, we made a comparison of TPH and TFL in cell suspension culture with leaf extracts. During culture time from *E. platycarpa* CSC, the TPH and TFL content had low variation (Figure 3).

During the adaptation phase of the culture, there was a notable increase in TPH (15.6 mg GAE/g DW) and then it remained constant, showing amounts between 5.2 ± 0.63 and 6.2 mg GAE/g DW. Similar studies reported by Giri et al. [28] found a concentration of 10.17 mg GAE/g DW after 28 days of culture of *Habenaria edgeworthii*. In this work, the TPH content of *E. platycarpa* leaves (11.77 mg GAE/g DW), was lower than cell suspension cultures on day 2; however, during this stage biomass production was low.

Regarding the TFL content, this also remained constant between days 2 and 14, except that day 0 had a value of 104.1 µg QE/g DW. Near the end of the senescence phase (day 16), the TFL decreased to 56.2 µg QE/g DW (Figure 3). On the other hand, low concentrations of TFL were also found in wild plant leaves (88.2 µg QE/g DW). In studies conducted in *Saussurea medusa* (Maxim) cell suspension cultures, the total flavonoid production was 607.8 mg/L after 15 days of culture [29]. In other species such as *Clinacanthus nutans*, large amounts of total polyphenols, phenolic acids, and flavonoids were found in in vitro plants compared with plants propagated by a stem cutting technique [6].

### 2.3. Yield of Extracts of Cell Suspension Culture and Intact Plant

In general, the methanolic and dichloromethane extracts exhibited the highest dry weight yields (Table 2). This coincides with what has been reported for related species such as *E. polystachya*, in which the highest yields were obtained with methanol and dichloromethane [30]. In *Tectona grandis*, yields of 2.9%, 2.3%, and 3.6% were reported, through a sequential extraction with hexane, ethyl acetate, and methanol, respectively [31]. The methanolic extracts made of the heartwood of other *Fabaceae* (*Caesalpinia platyloba* and *Lysiloma latisquum*) also report the best yields of extracts using these solvents [32]. For *E. platycarpa*, the highest yield of the methanolic extract was 36.9% of the heartwood and 30.08% for the leaf (Table 2). Other studies carried out on branches of *Severinia buxifolia* also obtained a higher extract yield (33.2%) using methanol [33]. In another study conducted on *Caesalpinia sappan* L., methanolic extracts of heartwood and leaf showed higher yields of 17.60% and 17.05%, respectively [34]. This may be because plants contain large amounts of polar compounds such as proteins and carbohydrates [35]. High amounts of polyols may also have increased the yield of methanolic extracts [36].

### 2.4. Compounds Identified by GC‒MS

The *Fabaceae* family produces a high diversity of bioactive compounds as defense against bacterial and fungi [37]. In the literature, few studies have reported on the determination of intact plant compounds of *E. platycarpa*; there are no studies on the phytochemical profile of its cell culture extracts.

By using gas chromatography‒mass spectrometry (GC‒MS) analysis, the phytochemical profile of the hexane, dichloromethane, and methanolic extracts of the cell suspension cultures, sapwood, heartwood, and leaves of the intact plant of *E. platycarpa* was determined (Table 3 and Table 4).

Chromatograms with the main compounds identified by GC‒MS are in Appendix A and their mass spectra are in Appendix A.

All hexane extracts of sapwood, heartwood, leaf, and CSC were composed mostly of saturated alkanoic acids (Saa), saturated diacids (Sd), and steroids (Ste), while only the leaves had triterpenes (Tri) (Figure 4A,B). The fatty and defatted hexane extracts of CSC stood out for producing saturated alkanoic acids (75%), mainly hexadecanoic acid (63%) (Table 3; Appendix A).

The largest amount of saturated diacids (nonanedioic acid) were found in the fatty hexane extract of sapwood (26.9%) and heartwood (15.3%) (Appendix A). On the other hand, high amounts of steroids were found in defatted hexane extracts of sapwood (77.2%) and heartwood (78.1%), mainly composed of *β*-sitosterol (~43%) and stigmasterol (~23%) (Table 3; Appendix A). It is possible that these compounds are found as natural derivatives in leaf extracts because only stigmasterol was found at a low abundance (1.55%), but the peaks at the retention times 52.16‒53.05 showed a complex fragmentation pattern that was not possible to characterize (Appendix A). Stigmasta-3,5-dien-7-one was also found in the fatty hexane extracts of sapwood and heartwood in amounts of 17.48% and 19.0%, respectively. Only the fatty and defatted hexane extracts from the leaves contained *β*-amyrin in amounts of 7.13% and 16.30%, respectively (Table 3; Appendix A).

In several studies, it has been shown that hexane extracts are mainly composed of fatty acids with hydrocarbons and traces of terpenes—for example, in the hexane extracts of *Anisopus mannii* leaves, 73.8% of compounds were identified, highlighting hexadecanoic acid (34%), hexadecyl oxirane (11%), and (*Z,Z,Z*)-9,12,15-octadecatrienoic acid (9.6%), which showed antimicrobial activity [38]. The hexane extracts of the heartwood of other *Fabaceae* such as *Robinia pseudoacacia* L. are also rich in hexadecanoic acid, trimethylsilyl ester (13.39%), (*Z,Z*)-9,12-octadecadienoic acid (10.10%), tetradecane (6.88%), bis-(2-ethylhexyl) phthalate (6.21%), and hexadecane (6.15%) [39].

In general, dichloromethane extracts of *E. platycarpa* were dominated mostly by saturated alkanoic acids and steroids, and unsaturated alkanoic acids (Uaa), and lesser amounts of aromatic compounds (Ac), saturated diacids (Sd), and sesquiterpenoids (Ses) (Figure 5A).

Dichloromethane extract of CSC showed a greater amount of saturated alkanoic acids (35.3% of hexadecanoic acid) and unsaturated alkanoic acids (13.8% of (*Z,Z*)-9,12-octadecadienoic acid) (Table 4 and Appendix A). The dichloromethane extracts of sapwood, heartwood, and CSC had similar amounts of steroids, from 13.5% to 14.7%. Only the leaves exhibited sesquiterpenoids (*β*-selinene, *γ*-muurolene, *β*-cadinene, 11-hydroxy-4*β*H,5*α*-eremophil-1(10)-ene and *trans,trans*-farnesol) in amounts < 1%. In contrast, all methanolic extracts exhibited polyol-type compounds (Figure 5B), mainly D-pinitol in the leaf extracts, with 74.3% (Table 4 and Appendix A). The rest of the compounds were mostly mono- and disaccharides, mainly in the CSC extracts

In the dichloromethane fractions of sapwood and heartwood extracts from *Quercus faginea*, saturated alkanoic acids have been found as the main constituents (15.7% and 25.8% in sapwood and heartwood, respectively), with hexadecanoic acid being the main compound (35.5% and 41.1%). In addition, sterols have been found between 10.2% to 13.0%, mainly *β*-sitosterol at 6.6% to 12.1% [40]. In a study, we found that in extracts with a mixture of chloroform and methanol (1:1) of *Cnidoscolus chayamansa* cell suspension cultures, lupeol acetate was obtained (38.1 mg/g extract), and the extracts had antibacterial activity [41]. Nonanedioic acid, *β*-sitosterol, stigmasterol, stigmasta-3,5-dien-7-one, *β*-amyrin, (*Z,Z*)-9,12-octadecadienoic acid, D-pinitol, hexadecanoic acid, and other compounds have been reported for *Eysenhardtia* genus and some *Fabaceae* species [10,11,42]; however, in this work, the phytochemical screening by GC‒MS analysis of plants and cell suspension culture extracts from *E. platycarpa* is reported for the first time.

### 2.5. Antifungal Activity

As an assay of biological activity, we evaluate the antifungal potential of hexanic (fatty and defatted), dichloromethane, and methanolic extracts from sapwood, heartwood, leaves, and cell suspension cultures on phytopathogenic fungi available in our laboratory (*Rhizoctonia solani* and *Sclerotium cepivorum*). These fungi have a broad host range and cause significant losses in terms of the yield and quality of many crop species. In a previous study, we reported the antifungal activity of extracts of *Eysenhardtia polystachya* [13], a species close to *E. platycarpa*. In the current work, it was found that the antifungal activity of *E. platycarpa* extracts was statistically significant (*p* = 0.05) on the mycelial growth of *R. solani* and *S. cepivorum* (Figure 6 and Figure 7).

Of the 16 extracts, only the fatty hexane and dichloromethane extracts showed better inhibition of both fungi species. The fatty hexane extract of sapwood had the maximum inhibition of mycelial growth for *R. solani* (51.4%), while the fatty hexane extract of cell suspension culture (CSC) showed 62.0% inhibition for *S. cepivorum* (Figure 6A). The other fatty hexane extracts had inhibition values lower than 25.0%. In the case of defatted hexane extracts, the inhibition percentages for *R. solani* were low (less than 22%), while the values for *S. cepivorum* ranged from 22% to 35% (Figure 6B). The fungicide Cercobin did not affect the growth of *R. solani* but was efficient at inhibiting the growth of *S. cepivorum*. In a previous study, we reported the antifungal activity of *E. polystachya* extracts, in which the defatted hexane extract of CSC showed 66.0% inhibition for *R. solani* and was also higher than Cercobin, while the fatty hexane extracts had low inhibition [13].

Regarding the dichloromethane extracts, only the cell suspension cultures extract showed moderate inhibition of *R. solani* (36.0%) compared with Cercobin (9.0%) (Figure 7A).

In contrast, the dichloromethane extract from the cell suspension culture showed the most effective inhibition against *S. cepivorum* (71.5%), followed by heartwood (55.2%) and leaf extracts (45.9%) (Figure 7A). In a study carried out on *E. polystachya*, we reported that dichloromethane extracts of sapwood and heartwood also inhibited the growth of *S. cepivorum* by 73.0% and 80.0%, respectively [13].

On the other hand, the methanolic extracts from *E. platycarpa* showed low inhibition compared with Cercobin. However, CSC extracts showed the maximum percentage of inhibition for *S. cepivorum*, 44.2% (Figure 7B).

Many plants have been reported to inhibit the in vitro growth of phytopathogenic fungi, which promise to be better than commercial fungicides [43,44]. It is possible that the growth inhibition of *R. solani* and *S. cepivorum* with the fatty hexane extract of the sapwood and CSC (Figure 6A) is due to a synergism between saturated fatty acids, saturated diacids, and steroids, since these are more abundant in this extract (Figure 4A and Table 3). In fact, the defatted hexanic extracts also contain saturated fatty acids and a high amount of steroids, but scarce saturated diacids compared with the fatty hexane extract. Therefore, there was also a decrease in the inhibition of fungal growth (Figure 6B).

Several free fatty acids (lauric acid, myristic acid, palmitic acid, oleic acid, and linoleic acid) are known to have an inhibitory effect on fungal germination, mycelial growth, and sporulation [45,46]. The possible mechanisms of antifungal activity have been studied previously and focused on fungal membrane disruption, causing an increase in membrane fluidity, causing leakage of the intracellular components and cell death [47] or interfering fungal sphingolipid biosynthesis [48]. In addition, they may influence the inhibition of protein and enzyme synthesis related to fatty acid metabolism [49]. On the other hand, the synergism of the aromatic compounds, along with the high amounts of phytosterols (stigmasterol and *β*-sitosterol), hexadecanoic acid, and unsaturated fatty acids (*Z,Z*)-9,12-octadecadienoic acid) found in CSC dichloromethane extract of *E. platycarpa* (Table 4) may have increased the inhibition for *S. cepivorum*. On the other hand, the CSC dichloromethane extract was the only one that significantly inhibited the growth for *R. solani*, perhaps due to the high amount of saturated and unsaturated alkanoic acids and steroids compared to the other dichloromethane extracts from intact plants (Figure 5A and Figure 7A).

The hexadecanoic acid (palmitic acid) from many plant species has been reported against phytopathogenic fungi, such as *Aspergillus niger*, *Botrytis cinerea*, *Colletotrichum lagenarium*, *Emericella nidulans*, *Fusarium oxysporum*, and *Alternaria solani* [45,49,50]. Phytosterols, e.g., stigmasterol and sitosterol, have also been reported to be effective against phytopathogenic fungi [51,52]. A mixture of stigmasterol and *β*-sitosterol isolated from the pericarp of *Areca catechu* markedly inhibited the spore germination, mycelial growth, and germ-tube elongation of *Colletotrichum gloeosporioides* [53]. Studies reported in other species of the *Fabaceae* family (*Tephrosia apollinea*, *Dahlstedtia glaziovii*, and *Deguelia duckeana*), have shown that dichloromethane extracts can inhibit fungal growth because they contain high amounts of flavonoids and prenylated phenolic compounds [26,54,55]. Therefore, some plant extracts may be a source of antifungal compounds since they have had to develop compounds to resist infections by fungi present in their environment [56].

## 3. Materials and Methods

### 3.1. Collection of Plant Material

The leaves, seeds, and the trunk from wild *Eysenhardtia platycarpa* plants were collected in November 2015, in San Luciano (Jocotepec, Estado de Jalisco, Mexico), located at 20°19′10.23″ N and 103°24′12″ W at an elevation of 1950 m a.s.l. A sample of the plant was used for identification, registered, and deposited in the Luz María Villarreal de Puga Herbarium, Instituto de Botanica, Universidad de Guadalajara (IBUG) with the voucher number 28112017.

### 3.2. Obtaining Plantlets and Incubation Conditions

All in vitro culture experiments were conducted at the Laboratorio de Cultivo In vitro de Plantas of the Departamento de Madera, Celulosa y Papel, of the Universidad de Guadalajara, México. First, uncoated seeds of *E. platycarpa* were washed with a soap solution for 10 min, followed by disinfection with 70% (*v*/*v*) ethanol for 30 s. Then, the seeds were disinfected with 1.2% (*v*/*v*) sodium hypochlorite solution (Cloralex^®,^ Industrias Alen, S.A. de C.V, Moterrey, Nuevo León, México) for 15 min along with four drops of Tween 20^®^ per 100 mL of disinfectant solution. After disinfection, the seeds were rinsed three times with sterile distilled water in a horizontal laminar flow cabinet (CFLH-90E, Novatech, Guadalajara, Jal. Mexico).

The disinfected seeds were sown in MS culture medium [57] supplemented with 3% sucrose (*w*/*v*) (Sigma-Aldrich, St. Louis, MO, USA); only half of the macronutrients were used. The culture medium was adjusted to pH 5.8 and then gelled with 2 g/L of Phytagel^®^ (Sigma-Aldrich). The culture medium was transferred to Gerber flasks of 100 mL capacity and sterilized in a manual autoclave (CV300-A, AESA, Tecamac, Estado de México, México) at 121 °C, 15 psi, for 18 min. Four disinfected seeds were placed in Gerber-type jars containing 25 mL of MS medium.

All cultures were incubated at 25 ± 2 °C under a 16-h photoperiod of white fluorescent light with a light intensity of 60 μmol/m^2^/s. Eight days after germination, the plantlets were transferred to a flask with a 1 L capacity, containing 80 mL of MS culture medium. Plantlets were used for subsequent experiments on callus induction.

### 3.3. Establishment of Callus Cultures

To induce calluses, 30-day-old plantlets grown in in vitro conditions were used as a source of explants. Internodal segments of approximately 1 cm in length were sown in Gerber flasks containing 25 mL of semisolid MS culture medium and plant growth regulators (PGRs). The PGRs consisted of two auxins, 2,4-dichlorophenoxyacetic acid (2,4-D) or naphthaleneacetic acid (NAA), each combined with kinetin (KIN) as a cytokinin; all PGRs were used at 0.0, 0.1, 1.0, and 2.0 mg/L. Each treatment consisted of four flasks with four explants per flask (*n* = 16), and the experiment was repeated twice. Cultures were maintained under the same incubation conditions as in seed germination and subcultured every four weeks with fresh culture medium.

### 3.4. Cell Suspension Cultures

#### 3.4.1. Growth Kinetics

The calluses that seemed more friable and showed better growth characteristics were used to initiate the establishment of the cell suspension cultures (CSC), using MS liquid medium with the same PGRs that induced the callus. Erlenmeyer flasks (125 mL capacity) containing 25 mL of culture medium were inoculated with 3 g of callus (FW). Cultures were incubated in an orbital shaker (PRENDO AGO-6040; Puebla, Pue. Mexico) at 110 rpm under a photoperiod (16 h light/8 h dark) of white fluorescent light with a light intensity of 60 μmol/m^2^/s and 25 ± 2 °C. When an increase of biomass was shown in the flasks, cells were harvested and filtered with 200-μm nylon mesh filters (Whatman No. 1) to remove excess culture medium. Then cells were subcultured every two weeks for six months in several flasks to increase the biomass, and no changes were observed in the culture. Kinetic growth was carried out in an Erlenmeyer flask (125 mL capacity) containing 25 mL of MS liquid medium and 1.5 g of fresh cells. The biomass contained in the three flasks was harvested every two days, filtered, and washed with distilled water to remove the excess culture medium. The biomass was dried in an oven at 50 °C until a constant weight. The experiment was repeated twice, and the biomass dry weight (DW) data were used to plot the growth curve. The value of the specific growth rate (*µ*) was calculated by plotting the natural logarithm of biomass versus time, between days 2 and 8 of growth culture (exponential phase). The slope of this linear part of the kinetics was defined as *μ* and is given in 1/unit of time (day^–1^). The generation (doubling) time (*td*) was calculated from the *μ* value and expressed as *td* (day) = ln(2)/*μ*, and the growth index was calculated considering the maximal biomass obtained with a reduction of the inoculum and divided by the inoculum. Biomass produced according to the sucrose content was determined based on the theoretical value (Y = g of maximum biomass/g of sucrose added to culture medium) reported for plants [20,23,58]. The modified Gompertz model is well known and widely used in many aspects of biology and has been frequently used to describe the plant growth [21,22]. The lag phase and other growth parameters were estimated with the modified Gompertz model [59] using Equation (1):(1)$$X_{t}=X_{max}\cdot \exp\left\{-\exp\left[\frac{\mu e}{X_{max}}\left(\lambda -t\right)+1\right]\right\}$$
where **X_t_** (g/L) is the biomass at any time *t* (days), *μ* (day^−1^) is the specific growth rate, *X_max_* (g/L) is the maximum cell growth achieved during the stationary phase, *λ* is the lag time (days), and e = 2.7182. The nonlinear regression was performed using the Kaleida Graph (4.0 Synergy Software, Reading, PA, USA) to predict the kinetic parameters.

After determining the growth kinetics, cells were subcultured every 12 days for three months. The accumulated biomass was washed with distilled water, dried at 40 °C, and stored frozen until subsequent experiments.

#### 3.4.2. Sucrose Consumption Determination

The culture medium filtered from the CSC of the growth kinetics was used to determine the total sugars. For each sampling, three flasks were harvested, and an aliquot of 5 mL was taken from each flask (*n* = 3) and stored frozen until analysis by the phenol–sulfuric method [60]. An aliquot of 250 μL of culture medium was diluted in distilled water (1:400); then, a 500-μL aliquot was taken and 500 μL of phenol (5%) was added; subsequently, 2.5 mL of concentrated sulfuric acid were added. The sample was vigorously mixed for 3 s and allowed to react at room temperature for 30 min. Samples were read in a spectrophotometer at 490 nm, using distilled water as a blank. To carry out the calibration curve, sucrose was used as a standard at concentrations of 1‒40 g/L.

#### 3.4.3. Determination of the Total Phenolics and Flavonoids Content

Biomass samples of CSC from growth kinetics and wild plant leaf were used to determine the total phenolics and flavonoids content. Samples of 50 mg of CSC or leaves were refluxed using 20 mL methanol (in water bath) at 65 °C for 20 min, and three extraction cycles were done for each sample. The extracts of the same samples were mixed, filtered, brought up to 20 mL, transferred to amber flasks, and stored frozen until analysis.

The total phenolics content (TPH) was quantified by the Folin‒Ciocalteu (FC) method [61]. An aliquot of 500 µL methanolic extract was mixed with 125 µL of FC and then 125 µL Na_2_CO_3_ (20% *w*/*v*) was added. The mixture was supplemented with distillated water up to a 2 mL total volume. The reaction was maintained at room temperature for 60 min in the dark. TPH was calculated based on the calibration curve of gallic acid at concentrations of 0 to 50 mg/L. Samples of methanolic extracts from leaves and CSC were analyzed in a Varian Cary^®^ 50 UV-Vis spectrophotometer (Agilent Technologies, Inc., Santa Clara, CA, USA) at 765 nm. The results were expressed in terms of gallic acid equivalents (GAE) in mg/g of dry biomass (DW).

The total flavonoids content (TFL) was quantified with the aluminum chloride colorimetric method [62]. An aliquot of methanolic extracts (240 μL) was mixed with 1.50 mL of distilled water, then 90 μL NaNO_2_ (5%) was added and allowed to react for 6 min in the dark. After the reaction, 180 μL AlCl_3_ (10%) was added to the mixture, which was stirred vigorously. After 5 min, 600 μL NaOH (1 M) was added and brought up to a 3 mL final volume with distilled water. Samples were analyzed in a Varian Cary^®^ 50 UV-Vis spectrophotometer at 510 nm. The calibration curve was performed with quercetin as the standard, using concentrations of 100 to 1600 μg/mL. The results were expressed in terms of quercetin equivalents (QE) in µg/g of dry biomass (DW).

### 3.5. Phytochemical Analysis of Cell Suspension Cultures and Intact Plants

#### 3.5.1. Extraction and Sample Preparation

The extraction process was carried out according to a previously reported methodology to obtain extracts or bioactive compounds from *E. platycarpa* or *E. polystachya* [12,13,63]. The biomass of the 12-day cell suspension cultures and samples from the intact plant (sapwood, heartwood, and leaves) were ground to a fine powder and dried in an oven at 50 °C. Separately, dried samples of sapwood (300 g), heartwood (269 g), leaves (238 g), and CSC (9.91 g) were extracted (four each) by maceration at room temperature with hexane, dichloromethane, and methanol for 72 h. After the extraction of each solvent, the samples were dried before adding the next solvent. The extracts were filtered and concentrated using a rotavapor BÜCHI EL-131 (BÜCHI Labortechnik AG, Flawil, Switzerland) and dried in an oven at 40 °C to remove traces of the solvent. Each hexane extract was extracted with methanol to obtain two extracts (a defatted hexane extract and a fatty hexane extract for each sample). The yield percentages were obtained with Equation (2):(2)$$Yield\;\left(\%\right)=\left(\frac{de}{db}\right)\times 100$$
where *de* is the dry weight of the extract and *db* is the dry weight of the biomass used for extraction.

A total of 16 dry extracts were obtained and derivatized with BSTFA to identify as many compounds as possible by GC‒MS analysis. The derivatization was carried out according to the methodology previously reported for extracts from *Quercus faginea* and *E. polystachya* [40,63]. Briefly, 2 mg of extracts were dissolved in 100 μL of pyridine and derivatized by adding 100 μL of bis(trimethylsilyl)trifluoroacetamide (BSTFA). The reaction mixture was heated to 50 °C and stirred for 30 min on a heating plate.

#### 3.5.2. Analysis of Extracts by Gas Chromatography‒Mass Spectrometry (GC‒MS)

The derivatized extracts (1 μL) were immediately injected into an Agilent 6890 instrument coupled to an Agilent Technologies 5973N Network Mass Selective Detector (GC‒MS) using electron impact as the ion source at 70 eV in a mass range of 20–600 DA (Agilent Technologies, Inc., Santa Clara, CA, USA). The capillary column was a HP-5MS (30 m × 0.25 mm, 0.25 μm film thickness; Agilent Technologies, Inc.). The oven temperature was initially set to 100 °C (for 1 min) and rose at a rate of 10 °C/min up to 150 °C, then at a rate of 3 °C/min up to 300 °C (for 4 min). Helium was used as the carrier gas with a flow rate of 1.0 mL/min, and the injector temperature was set at 250 °C. Compounds were identified as TMS derivatives by comparing their mass spectra with the NIST library version 1.7a and by comparing their fragmentation patterns with published data [63,64]. For determining the relative percentage amounts, peaks were integrated using a GC ChemStation software version C.00.01. The composition was reported as a percentage of the total peak area.

### 3.6. In Vitro Antifungal Evaluation of Extracts

The phytopathogenic fungi *R. solani* and *S. cepivorum* were provided by the Colección del Laboratorio de Patología, Departamento de Producción Agrícola (Universidad de Guadalajara). Seven-day-old strains grown on potato dextrose agar (PDA, BD Bioxon; Ciudad de Mexico, Mexico) culture medium was used. The antifungal evaluation was carried out according to the agar disk-diffusion method [65]. The 16 extracts were dissolved in 96% ethanol (1 mg/mL) and the 5 mm diameter filter paper discs (Whatman No. 1) superposed on PDA culture medium were impregnated with 10 μL of each solution. Cercobin^®^ and ethanol were used as positive and negative controls, respectively. After applying solution extracts or controls on the discs, the solvent was allowed to evaporate for 1 h in a laminar flow cabinet. Mycelium propagules (5 mm^3^) were inoculated on discs treated into sterile polystyrene Petri dishes (Interlux^®^ 90 × 15 mm) containing 10 mL of PDA medium and then incubated at 28 ± 2 °C. The mean radial mycelial growth was determined by measuring the colony diameter 72 h after inoculation. The mean growth values were obtained and then converted into the inhibition percentage of mycelial growth (*GI*) in relation to the control treatments using Equation (3) [65]:(3)$$GI=\left(\%\right)=\left(\frac{dc-d_{t}}{dc}\right)\times 100,\;$$
where *dc* is the average diameter of the fungal colony with control and *dt* the average diameter of fungal colony with treatments.

### 3.7. Statistical Analysis

The data corresponding to the percentage of callus induction, dry biomass of cell suspension cultures, total phenolics and flavonoids content, and the inhibition percentage of the mycelial growth were subjected to a normality test and then an analysis of variance (ANOVA), followed by Tukey’s multiple range test (*p* = 0.05). SAS 9.0 software (SAS Institute, Inc., Cary, NC, USA) was used for the statistical analysis. All experiments were conducted in triplicate.

## 4. Conclusions

In this work, the obtaining of a biotechnological culture of *E. platycarpa*, the phytochemical profile of cell cultures and intact plant, and its antifungal activity are reported for the first time. Although the callus induction response with auxins 2,4-D and NAA in most of the treatments was greater than 80%, NAA was the most efficient auxin to establish the cell suspension culture. Cell cultures were able to produce some compounds found in wild plants, and in other cases, they produced different or larger amounts of compounds. The dichloromethane extract of CSC showed greater effectiveness at inhibiting the in vitro growth of *S. cepivorum*, and among these extracts, the CSC were the only ones that inhibited the growth of *R. solani*. Thus, these extracts could be a sustainable alternative resource to synthetic fungicides. Nevertheless, in future studies, research can be performed to increase the production of compounds and antifungal activity.

## Figures and Tables

**Figure 1 plants-10-00414-f001:**
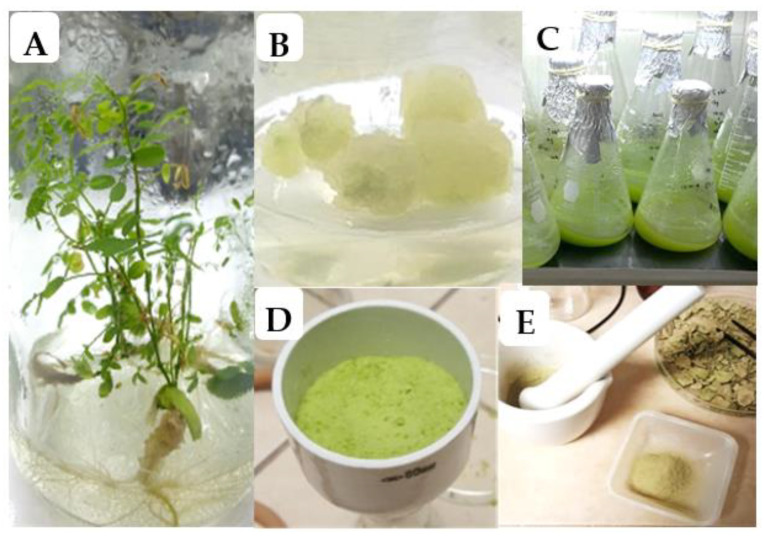
In vitro cultures of *Eysenhardtia platycarpa*. (**A**) Plantlets grown in MS culture medium without plant growth regulators; (**B**) callus production at 15 days of culture; (**C**) cell suspension cultures with 2 mg/L NAA and 0.5 mg/L KIN; (**D**) fresh biomass harvested after 12 days of culture; (**E**) dried biomass used for obtaining extracts.

**Figure 2 plants-10-00414-f002:**
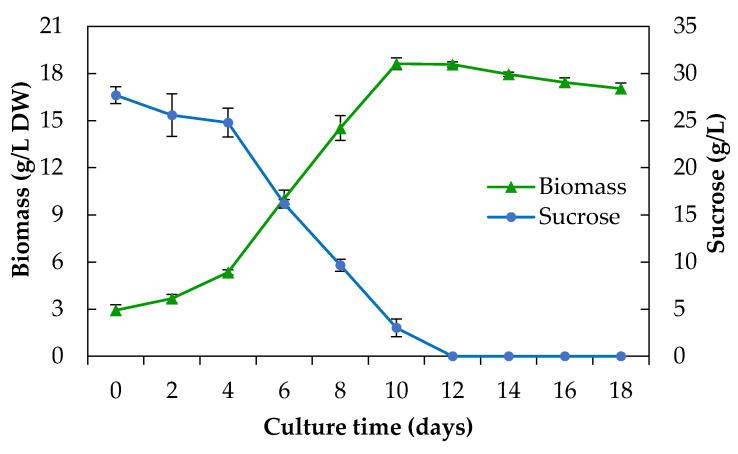
Growth kinetics and consumption of sucrose of a cell suspension culture of *Eysenhardtia platycarpa* for 18 days of culture. Values represent the mean ± standard deviation of three replicates.

**Figure 3 plants-10-00414-f003:**
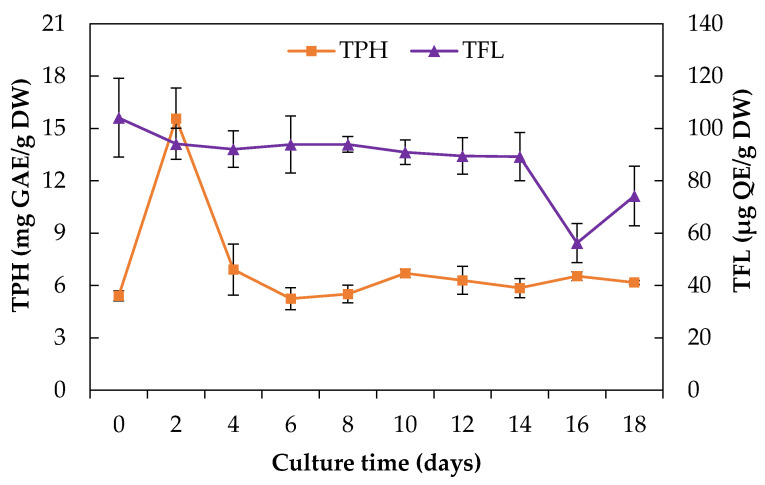
Production of total phenolics (TPH) and total flavonoids content (TFL) of a cell suspension culture of *Eysenhardtia platycarpa* over 18 days of culture. Values represent the mean ± standard deviation of three replicates.

**Figure 4 plants-10-00414-f004:**
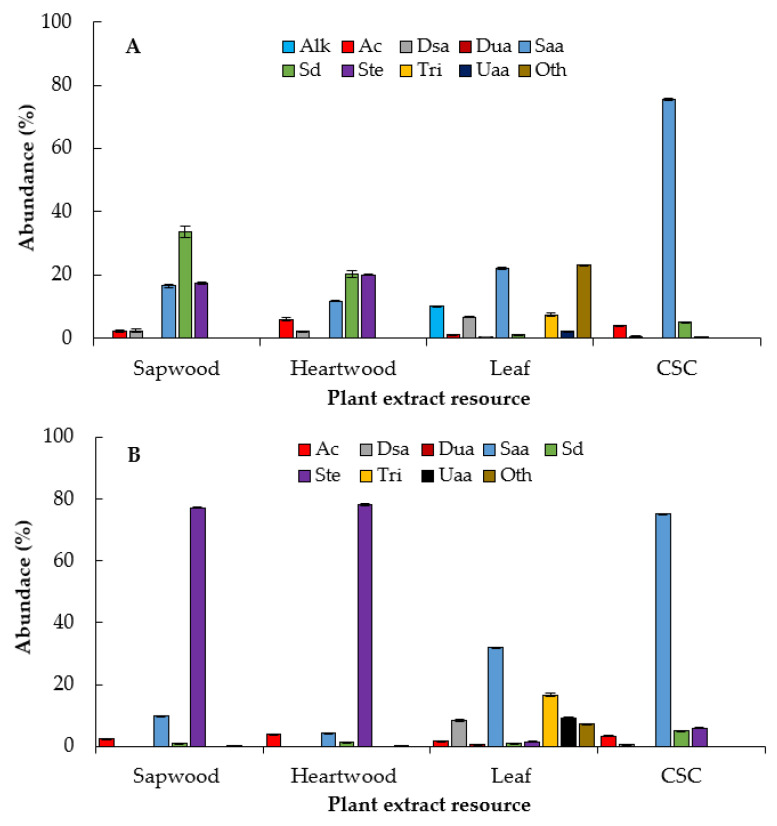
Chemical groups summarized by plant extracts resource of *E. platycarpa*. (**A**) Fatty hexane extract; (**B**) defatted hexane extract. Alk: alkanes; Ac: aromatic compounds; Dsa: derivatives from saturated alkanoic acids; Dua: derivatives from unsaturated alkanoic acids; Saa: saturated alkanoic acids; Sd: saturated diacids; Ste: steroids; Tri: triterpenoids; Uaa: unsaturated alkanoic acids; Oth: others. CSC: Cell suspension cultures. Values represent the mean ± standard deviation of two replicates.

**Figure 5 plants-10-00414-f005:**
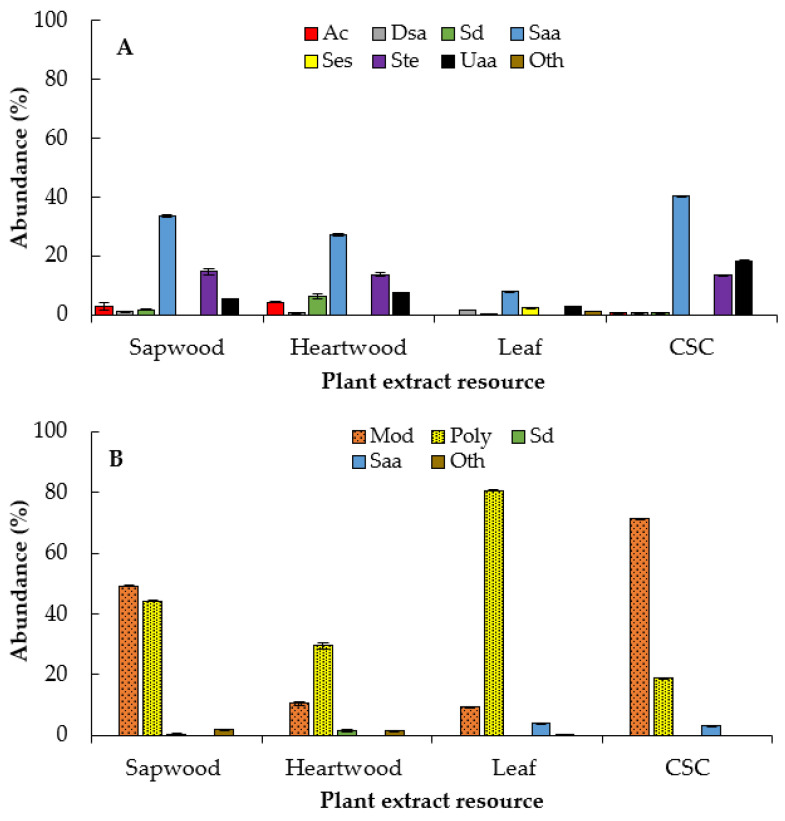
Chemical groups of *E. platycarpa* extracts, summarized by plant extract source. (**A**) Dichloromethane extract; (**B**) methanolic extract. Ac: aromatic compounds; Dsa: derivatives from saturated alkanoic acids; Mod: mono and disaccharides; Poly: polyols; Sd: saturated diacids; Saa: saturated alkanoic acids; Ses: sesquiterpenoids; Ste: steroids; Uaa: unsaturated alkanoic acids; Oth: others. CSC: Cell suspension cultures. Values represent the mean ± standard deviation of two replicates.

**Figure 6 plants-10-00414-f006:**
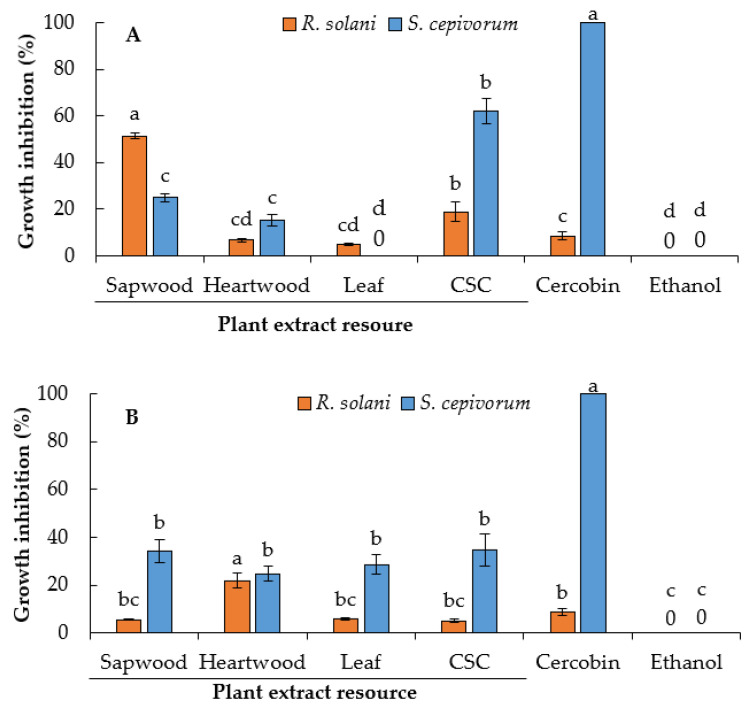
Effect of different source extracts of *Eysenhardtia platycarpa* on inhibition of mycelial growth of *Rhizoctonia solani* and *Sclerotium cepivorum* after 72 h of culture. (**A**) Fatty hexane extract; (**B**) defatted hexane extract. CSC: Cell suspension culture. Values represent mean ± standard deviation of three replicates. Bars followed by the same letter between the same fungus are not significantly different (*p* = 0.05) using Tukey’s multiple range test.

**Figure 7 plants-10-00414-f007:**
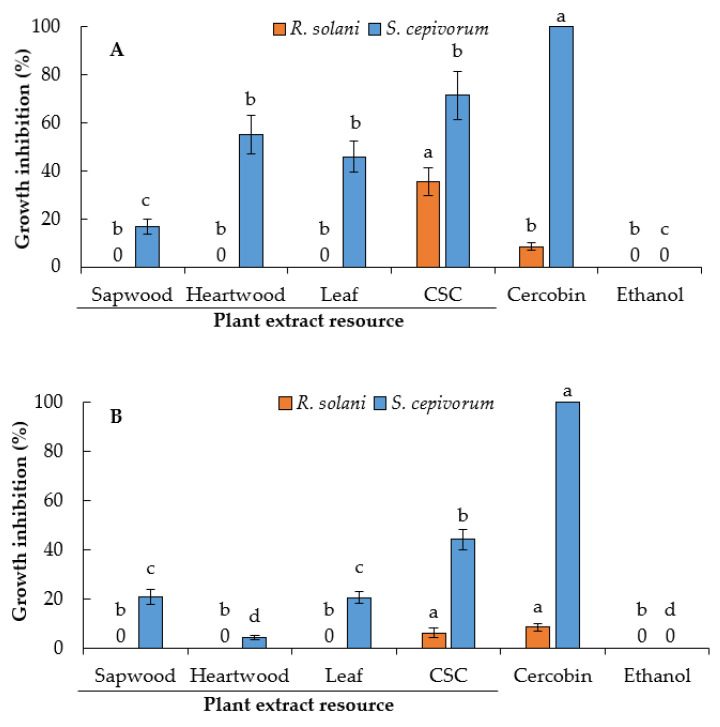
Effect of different source extracts of *Eysenhardtia platycarpa* on the inhibition of mycelial growth of *Rhizoctonia solani* and *Sclerotium cepivorum* at 72 h of culture: (**A**) Dichloromethane extract; (**B**) methanolic extract. CSC: Cell suspension culture. Values represent mean ± standard deviation of three replicates. Bars followed by the same letter between the same fungus are not significantly different (*p* = 0.05) using Tukey’s multiple range test.

**Table 1 plants-10-00414-t001:** Percentage of internodal explants of *Eysenhardtia platycarpa* inducing callus in Murashigue and Skoog (MS) culture medium after 30 days of culture.

PGRs (mg/L)		Callus Induction (%)	PGRs (mg/L)		Callus Induction (%)
2,4-D	KIN		NAA	KIN	
0.0	0.0	50.0 ± 0.0 ^b^	0.0	0.0	50 ± 0.0 ^c^
0.0	0.1	66.7 ± 7.2 ^ab^	0.0	0.1	58.3 ± 19.1 ^bc^
0.0	1.0	93.8 ± 6.3 ^a^	0.0	1.0	100.0 ± 0.0 ^a^
0.0	2.0	75.0 ± 10.2 ^ab^	0.0	2.0	100.0 ± 0.0 ^a^
0.1	0.0	81.3 ± 12.0 ^ab^	0.1	0.0	75.0 ± 10.2 ^ab^
0.1	0.1	81.3 ± 12.0 ^ab^	0.1	0.1	93.8 ± 6.3 ^a^
0.1	1.0	93.8 ± 6.3 ^a^	0.1	1.0	93.8 ± 6.3 ^a^
0.1	2.0	87.5 ± 12.5 ^ab^	0.1	2.0	100.0 ± 0.0 ^a^
1.0	0.0	75.0 ± 10.2 ^ab^	1.0	0.0	93.8 ± 6.3 ^a^
1.0	0.1	100.0 ± 0.0 ^a^	1.0	0.1	100.0 ± 0.0 ^a^
1.0	1.0	83.3 ± 14.4 ^ab^	1.0	1.0	100.0 ± 0.0 ^a^
1.0	2.0	100.0 ± 0.0 ^a^	1.0	2.0	100.0 ± 0.0 ^a^
2.0	0.0	83.3 ± 14.4 ^ab^	2.0	0.0	100.0 ± 0.0 ^a^
2.0	0.1	100.0 ± 0.0 ^a^	2.0	0.1	100.0 ± 0.0 ^a^
2.0	1.0	91.7 ± 7.2 ^a^	2.0	1.0	100.0 ± 0.0 ^a^
2.0	2.0	100.0 ± 0.0 ^a^	2.0	2.0	100.0 ± 0.0 ^a^

PGRs: Plant growth regulators; 2,4-D: 2,4‒dichlorophenoxyacetic acid; NAA: naphthaleneacetic acid; KIN: kinetin. Values represent mean ± standard deviation of four replicates per treatment in two repeated experiments. Means followed by the same letter in superscript in the same column are not significantly different (*p* = 0.05) according to Tukey’s multiple range test.

**Table 2 plants-10-00414-t002:** Yields of different solvent extraction of intact plant and cell suspension culture of *Eysenhardtia platycarpa*.

Extract	Yield of Extract (%)			
	Sapwood	Heartwood	Leaf	Cell Suspension Culture
Fatty hexane	4.0	6.7	8.0	0.1
Defatted hexane	5.0	8.6	10.1	0.5
Dichloromethane	15.0	22.8	20.0	2.4
Methanolic	21.1	36.9	30.8	23.8
**Total**	45.1	75.0	68.9	26.8

**Table 3 plants-10-00414-t003:** Chemical constituents of the hexane extracts of cell suspension culture (CSC) and plant of *Eysenhardtia platycarpa* identified by GC‒MS.

Compound Name	Chemical Formula	RT (min)	Abundance (%)							
			Fatty Hexane Extract				Defatted Hexane Extract			
			Sapwood	Heartwood	Leaf	CSC	Sapwood	Heartwood	Leaf	CSC
**Alkanes**										
Heptacosane	C_27_H_56_	38.38	-	-	2.46 ± 0.02	-	-	-	-	-
Nonacosane	C_29_H_60_	42.98	-	-	6.36 ± 0.01	-	-	-	-	-
Octacosane	C_28_H_58_	47.26	-	-	1.31 ± 0.02	-	-	-	-	-
**Aromatic compounds**										
Isophthalic acid	C_8_H_6_O_4_	13.9	1.19 ± 0.09	-	-	-	-	-	-	-
Terephthalic acid	C_8_H_6_O_4_	14.72	1.30 ± 0.23	-	-	-	-	-	-	-
Dibutyl phthalate	C_16_H_22_O_4_	19.14	-	5.49 ± 0.60	0.69 ± 0.06	3.91 ± 0.03	2.51 ± 0.05	3.91 ± 0.01	1.05 ± 0.05	2.48 ± 0.13
1,2-Dihydroxyanthraquinone	C_14_H_8_O_4_	31.87	-	-	-	-	-	-	0.28 ± 0.04	-
Bis(2-ethylhexyl) phthalate	C_24_H_38_O_4_	34.69	-	-	-	-	-	-	-	0.79 ± 0.02
4,5-Dihydroxyanthraquinone-2-carboxylic acid	C_15_H_8_O_6_	41.37	-	-	0.47 ± 0.05	-	-	-	-	-
**Derivatives from saturated alkanoic acids**										
2-Hydroxyheptanoic acid	C_7_H_14_O_3_	6.41	1.23 ± 0.50	1.40 ± 0.05	-	-	-	-	-	-
3-Phenylpropanoic acid	C_9_H_10_O_2_	7.12	-	-	4.38 ± 0.09	-	-	-	6.28 ± 0.33	-
3-Hydroxyoctanoic acid	C_8_H_16_O_3_	7.98	0.78 ± 0.06	0.80 ± 0.02	-	-	-	-	-	-
3-(4-Methoxyphenyl) propionic acid	C_10_H_12_O_3_	11.49	-	-	2.34 ± 0.11	-	-	-	2.48 ± 0.13	-
2,3-Dihydroxypropyl palmitate	C_19_H_38_O_4_	36.21	-	-	-	0.60 ± 0.04	-	-	-	0.53 ± 0.00
**Derivatives from unsaturated alkanoic acids**										
3-Phenylprop-2-enoic acid	C_9_H_8_O_2_	9.2	-	-	0.55 ± 0.02	-	-	-	0.76 ± 0.06	-
**Saturated alkanoic acids**										
Tetradecanoic acid	C_14_H_28_O_2_	16.07	-	-	1.06 ± 0.02	0.39 ± 0.00	0.20 ± 0.01	-	1.56 ± 0.07	0.42 ± 0.00
Pentadecanoic acid	C_15_H_30_O_2_	18.74	-	-	-	0.42 ± 0.01	-	-	-	0.33 ± 0.03
Hexadecanoic acid	C_16_H_32_O_2_	21.49	14.32 ± 0.45	10.06 ± 0.13	11.48 ± 0.12	62.58 ± 0.36	6.25 ± 0.08	3.37 ± 0.03	19.02 ± 0.02	63.01 ± 0.07
Heptadecanoic acid	C_17_H_34_O_2_	24.22	-	-	-	0.85 ± 0.01	0.32 ± 0.00	0.17 ± 0.02	-	0.80 ± 0.00
Octadecanoic acid	C_18_H_36_O_2_	26.93	1.90 ± 0.09	1.49 ± 0.07	2.93 ± 0.08	6.09 ± 0.00	3.11 ± 0.05	0.70 ± 0.01	5.14 ± 0.10	6.91 ± 0.06
Eicosanoic acid	C_20_H_40_O_2_	32.17	-	-	0.55 ± 0.00	1.14 ± 0.00	-	0.14 ± 0.01	0.92 ± 0.03	1.00 ± 0.01
**Saturated alkanoic acids**										
Docosanoic acid	C_22_H_44_O_2_	37.08	-	-	1.19 ± 0.01	0.84 ± 0.00	-	-	1.51 ± 0.05	0.61 ± 0.01
Tetracosanoic acid	C_24_H_48_O_2_	41.71	-	-	5.09 ± 0.03	2.81 ± 0.01	-	-	3.95 ± 0.05	2.15 ± 0.05
Saturated diacids										
Hexanedioic acid	C_6_H_10_O_4_	8.49	0.91 ± 0.01	1.79 ± 0.53	-	-	-	0.19 ± 0.04	-	-
Heptanedioic acid	C_7_H_12_O_4_	10.33	1.54 ± 0.02	-	-	-	-	-	-	-
Octanedioic acid	C_8_H_14_O_4_	12.44	5.41 ± 0.36	2.90 ± 0.34	-	0.96 ± 0.09	0.18 ± 0.00	0.21 ± 0.02	-	0.82 ± 0.05
Nonanedioic acid	C_9_H_16_O_4_	14.87	27.93 ± 1.48	16.22 ± 1.27	1.02 ± 0.02	4.18 ± 0.03	0.94 ± 0.04	1.03 ± 0.01	0.92 ± 0.05	3.97 ± 0.10
**Steroids**										
Campesterol	C_28_H_48_O	50.45	-	-	-	-	8.76 ± 0.07	9.15 ± 0.20	-	-
Stigmasterol	C_29_H_48_O	51.14	-	1.19 ± 0.14	-	0.55 ± 0.08	23.56 ± 0.19	23.50 ± 0.11	1.55 ± 0.02	2.58 ± 0.08
*β*-Sitosterol	C_29_H_50_O	52.32	-	-	-	-	43.76 ± 0.03	43.43 ± 0.14	-	1.87 ± 0.01
Stigmastanol	C_29_H_52_	52.43	-	-	-	-	1.30 ± 0.12	1.04 ± 0.08	-	1.42 ± 0.02
Stigmasta-3,5-dien-7-one	C_29_H_46_O	53.27	17.48 ± 0.35	19.00 ± 0.29	-	-	-	-	-	-
Sitostenone	C_29_H_48_O	54.05	-	-	-	-	-	0.57 ± 0.04	-	-
**Triterpenoids**										
*β*-Amyrin	C_30_H_50_O	52.16	-	-	7.13 ± 0.46	-	-	-	16.30 ± 0.49	-
**Unsaturated alkanoic acids**										
(*Z*,*Z*)-9,12-Octadecadienoic acid	C_18_H_32_O_2_	25.99	-	-	0.67 ± 0.01	-	-	-	2.46 ± 0.06	-
(*Z*,*Z*,*Z*)-9,12,15-Octadecatrienoic acid	C_18_H_30_O_2_	26.17	-	-	1.49 ± 0.01	-	-	-	6.88 ± 0.16	-
(*Z*)-9-Octadecenoic acid	C_18_H_34_O_2_	26.18	-	-	-	-	0.25 ± 0.01	0.25 ± 0.01	-	-
**Others**										
6,10,14-Trimethylpentadecan-2-one	C_18_H_36_O	15.91	-	-	0.53 ± 0.03	-	-	-	0.66 ± 0.03	-
3,7,11,15-Tetramethyl-2-hexadecen-1-ol	C_20_H_40_O	25.11	-	-	-	-	-	-	2.89 ± 0.15	-
Octacosanol	C_28_H_58_O	48.27	-	-	22.68 ± 0.01	-	-	-	3.85 ± 0.13	-

Values represent the mean ± standard deviation of two replicates.

**Table 4 plants-10-00414-t004:** Chemical constituents of the Dichloromethane and methanolic extracts of cell suspension culture (CSC) and plant of *Eysenhardtia platycarpa* identified by GC‒MS.

Compound Name	Chemical Formula	RT(min)	Abundance (%)							
			Dichloromethane Extract				Methanolic Extract			
			Sapwood	Heartwood	Leaf	CSC	Sapwood	Heartwood	Leaf	CSC
**Aromatic compounds**										
Vanillin	C_8_H_8_O_3_	9.09	0.53 ± 0.32	-	-	-	-	-	-	-
Terephthalic acid	C_8_H_6_O_4_	14.72	0.39 ± 0.12	-	-	-	-	-	-	-
Dibutyl phthalate	C_16_H_22_O_4_	19.14	3.05 ± 0.82	2.97 ± 0.16	-	0.56 ± 0.01	-	-	-	-
1,2-Dihydroxyanthraquinone	C_14_H_8_O_4_	31.87	-	1.58 ± 0.01	-	0.43 ± 0.03	-	-	-	-
**Derivatives from saturated alkanoic acids**										
3-Phenylpropanoic acid	C_9_H_10_O_2_	7.12	-	-	1.02 ± 0.00	-	-	-	-	-
3-(4-Methoxyphenyl) propionic acid	C_10_H_12_O_3_	11.49	-	-	0.65 ± 0.02	-	-	-	-	-
2,3-Dihydroxypropyl palmitate	C_19_H_38_O_4_	36.21	1.19 ± 0.00	0.72 ± 0.01	-	0.96 ± 0.01	-	-	-	-
**Mono and disaccharides**										
Ketohexoses	C_6_H_12_O_6_	*	-	-	-	-	10.12 ± 0.10	3.14 ± 0.08	3.61 ± 0.02	27.77 ± 0.03
Aldohexoses	C_6_H_12_O_6_	*	-	-	-	-	6.30 ± 0.32	3.49 ± 0.24	2.88 ± 0.07	2.79 ± 0.00
Furanoses	C_6_H_12_O_6_	*	-	-	-	-	4.67 ± 0.08	-	0.96 ± 0.02	-
Disaccharides (glucose with fructose)	C_12_H_22_O_11_	*	-	-	-	-	28.24 ± 0.44	3.55 ± 0.12	1.82 ± 0.00	40.91 ± 0.09
**Polyols**										
L-threitol	C_4_H_10_O_4_	8.77	-	-	-	-	-	-	-	0.42 ± 0.00
Xylitol	C_5_H_12_O_5_	13.52	-	-	-	-	0.40 ± 0.00	4.23 ± 0.15	-	2.12 ± 0.00
D-pinitol	C_7_H_14_O_6_	16.38	-	-	-	-	37.00 ± 0.22	15.56 ± 0.47	74.28 ± 0.01	-
Ribitol	C_5_H_12_O_5_	18.11	-	-	-	-	-	-	-	9.22 ± 0.01
Sorbitol	C_6_H_14_O_6_	19.21	-	-	-	-	-	2.07 ± 0.08	-	1.84 ± 0.00
Inositol	C_6_H_12_O_6_	*	-	-	-	-	7.07 ± 0.05	7.00 ± 0.22	6.46 ± 0.02	5.22 ± 0.01
**Saturated diacids**										
Hexanedioic acid	C_6_H_10_O_4_	8.49	-	0.48 ± 0.02	-	-	-	-	-	-
Octanedioic acid	C_8_H_14_O_4_	12.44	-	0.74 ± 0.10	-	-	-	-	-	-
Nonanedioic acid	C_9_H_16_O_4_	14.87	1.94 ± 0.09	4.53 ± 0.73	0.51 ± 0.01	0.63 ± 0.01	0.44 ± 0.09	1.17 ± 0.49	-	-
**Saturated alkanoic acids**										
Tetradecanoic acid	C_14_H_28_O_2_	16.07	-	-	0.36 ± 0.01	-	-	-	-	-
Hexadecanoic acid	C_16_H_32_O_2_	21.49	27.42 ± 0.26	22.40 ± 0.26	5.20 ± 0.00	35.02 ± 0.44	-	-	2.06 ± 0.00	2.05 ± 0.00
Heptadecanoic acid	C_17_H_34_O_2_	24.22	0.77 ± 0.13	0.87 ± 0.02	-	0.36 ± 0.00	-	-	-	-
Octadecanoic acid	C_18_H_36_O_2_	26.93	3.37 ± 0.21	3.21 ± 0.04	1.44 ± 0.00	2.75 ± 0.22	-	-	1.88 ± 0.01	1.16 ± 0.03
Eicosanoic acid	C_20_H_40_O_2_	32.17	0.58 ± 0.04	-	-	0.42 ± 0.00	-	-	-	-
Docosanoic acid	C_22_H_44_O_2_	37.08	0.54 ± 0.01	-	-	0.29 ± 0.01	-	-	-	-
Tetracosanoic acid	C_24_H_48_O_2_	41.71	1.25 ± 0.10	0.90 ± 0.00	0.96 ± 0.03	1.27 ± 0.01	-	-	-	-
**Sesquiterpenoids**										
*β*-Selinene	C_15_H_24_	8.31	-	-	0.46 ± 0.01	-	-	-	-	-
*γ*-Muurolene	C_15_H_24_	8.74	-	-	0.32 ± 0.01	-	-	-	-	-
*β*-Cadinene	C_15_H_24_	11.2	-	-	0.77 ± 0.03	-	-	-	-	-
11-Hydroxy-4*β*H,5*α*-eremophil-1(10)-ene	C_15_H_26_O	13.42	-	-	0.60 ± 0.02	-	-	-	-	-
Trans, trans-farnesol	C_15_H_26_O	15.07	-	-	0.49 ± 0.01	-	-	-	-	-
**Steroids**										
Campesterol	C_28_H_48_O	50.45	1.61 ± 0.11	1.76 ± 0.11	-	0.82 ± 0.11	-	-	-	-
Stigmasterol	C_29_H_48_O	51.14	4.06 ± 0.41	3.96 ± 0.16	-	6.72 ± 0.05	-	-	-	-
*β*-Sitosterol	C_29_H_50_O	52.32	8.19 ± 0.63	7.50 ± 0.55	-	6.00 ± 0.07	-	-	-	-
**Unsaturated alkanoic acids**										
(*Z*,*Z*)-9,12-Octadecadienoic acid	C_18_H_32_O_2_	25.99	3.99 ± 0.06	5.20 ± 0.05	0.72 ± 0.03	13.63 ± 0.16	-	-	-	-
(*Z*,*Z*,*Z*)-9,12,15-Octadecatrienoic acid,	C_18_H_30_O_2_	26.17	-	-	2.29 ± 0.00	3.79 ± 0.06	-	-	-	-
(*Z*)-9-Octadecenoic acid	C_18_H_34_O_2_	26.18	1.62 ± 0.04	2.49 ± 0.05	-	0.82 ± 0.01	-	-	-	-
**Others**										
Malic acid	C_4_H_6_O_5_	8.3	-	-	-	-	1.95 ± 0.01	-	-	-
L-Threonic acid	C_4_H_8_O_5_	9.79	-	-	-	-	-	-	0.42 ± 0.00	-
Galactaric acid	C_6_H_10_O_8_	20.16	-	-	-	-	-	1.68 ± 0.06	-	-
3,7,11,15-Tetramethyl-2-hexadecen-1-ol	C_20_H_40_O	25.11	-	-	1.06 ± 0.00	-	-	-	-	-

Values represent the mean ± standard deviation of two replicates. * Several stereoisomers were found at different retention times.

## Data Availability

The data presented in this study are available on request from the corresponding author. The data are not publicly available due to privacy.

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
