# Peer review of "Establishment of a Cell Suspension Culture of Eysenhardtia platycarpa: Phytochemical Screening of Extracts and Evaluation of Antifungal Activity"

_plants, 2021, doi:10.3390/plants10020414_

Round 1

Reviewer 1 Report

The authors practically did not revise the manuscript - they corrected the erroneous results, and they brought standard methods for calculating growth characteristics - which is clearly unnecessary in the experimental article.

At the same time, the authors paid very serious attention to the responses to the comments of the reviewer, mainly trying to prove their incorrectness.

The reviewer does not see the need to enter into polemics with the authors - reviews are prepared to evaluate the manuscript and improve it. But a number of the authors' answers are at least surprising. For example "Light is an essential factor in the growth of cell cultures" "Only in specific cases, darkness can be used, but not during all culture growth". Everything is exactly the opposite - plant cell cultures are heterotrophic cells, and are usually grown in the dark. Light is used only when growing mycostrophic or autotrophic cell cultures. Various variants of the regulatory action of light are possible, but these are special research tasks. At the same time, for cultures of producing cells, the need for light regulation is rather a minus than a plus - how can this be done in a bioreactor?

The authors explained the method of maintaining the cell culture (however, very confusingly) - "the initial inoculum size for a 1 L total volume flask is 12 g contained in 200 mL (5%)", that is, it can be understood that a one-liter flask with 200 ml of medium was introduced 10 ml of inoculum with 0.6 g of cells. - however, in the manuscript the growing conditions are completely different "Kinetic growth was carried out in Erlenmeyer flask of 125 mL containing 25 mL of MS liquid medium and 1.5 g of cells."  Firstly, the degree of aeration of the culture decreases with a decrease in the volume of flasks and the volume of the suspension should be reduced, and secondly, what does"1.5g of cell" mean? Is this fhesh weight? Then, based on the graph, the DW FW ratio is 1: 20 (1.5 g in 25 ml, that is, in 1000 ml - 60 g, the initial density according to the graph is 3 g / l) and in the hospital the fresh weight at the time of inoculation (18 g by dry weight) should be 360 g / l, which is practically impossible.   This question was in the first review - and the authors decided not to answer it. By the way, this is a good illustration of the benefits of determining fresh weight: The opinion of the authors of "Measuring the dry weight is more suitable than the fresh weight, since there is a lot of variation and the results in fresh weight are not correct due to the variation in moisture content." is certainly true, and it necessary to measure both dry weight and fresh weigh  at least to avoid obvious mistakes .

An accurate calculation of growth parameters is certainly useful, but if this is done for only one growth curve, it makes little sense since the curve is shown. These characteristics - and most importantly, their spread - are important for assessing the stability of crop growth when analyzing at least 3 - 4 curves.

The authors believe that “GC – MS is a classical technique used in natural product research to separate and identify volatile compounds (e. G. Essential Oils). Thus, in the case of extracts with non-volatile compounds, the alkylsilyl derivatives have found extensive use for imparting high volatility and stability to the compounds for GC – MS analysis ”As for essential oils, the authors are absolutely right, but about the rest (despite the fact that The reviewer respects GC-MS very much and worked for a long time on the first GC-MS LKB GC-MS 2091 device ..) unfortunately not - at least from the molecular weight limit, not to mention possible artifacts when obtaining derivatives .. It is obvious that now HPLC-MS is the classic method of such work. For GC – MS analysis, there are clear boundaries of application where it is adequate.

It makes no sense to discuss the authors' comments on the methodology and principles of the chemical part of the work, the main points were said in the first review. The only strange logic is “About of the phytosterols, we included them because their antifungal activity has been reported, eg, stigmasterol and sitosterol have also been reported to be effective against phytopathogenic fungi” .. That is, the choice of analytes is based on their antimicrobial activity and about the rest of the analyzed compounds have data?

In conclusion, the analysis was carried out for only one crop cultivation and only once. Are there any guarantees that in another cultivation cycle the composition and amount of the analyzed substances will not be different - and, accordingly, their biological activity? Similarly - is there any guarantee that the main antimicrobial activity is not provided by those compounds that cannot be determined by GC MS?

Unfortunately,  the authors made only minor changes to the text of the manuscript, and   publication in Plants does not  recommended.

Author Response

Response to Reviewer 1 Comments

Point 1: The authors practically did not revise the manuscript - they corrected the erroneous results, and they brought standard methods for calculating growth characteristics - which is clearly unnecessary in the experimental article.

Response 1: We thoroughly reviewed and improved the entire manuscript and improved some mistakes that most reviewers noted. The earlier manuscript contained a widely used method for calculating growth parameters. In a previous comment, it was questioned and doubted how some parameters were calculated, so we decided to add another method (modified Gompertz model) to verify that the results of the parameters are correct, and it was. This can be verified in the manuscript in the Results and Methodology section. Page 4, lines 131-139; Pages 16 and 17, lines 412-426

Point 2: At the same time, the authors paid very serious attention to the responses to the comments of the reviewer, mainly trying to prove their incorrectness.

Response 1: We carefully reviewed and improved the entire manuscript based on all reviewers' observations and comments and it was corrected where necessary.

Point 3: The reviewer does not see the need to enter into polemics with the authors - reviews are prepared to evaluate the manuscript and improve it. But a number of the authors' answers are at least surprising. For example "Light is an essential factor in the growth of cell cultures" "Only in specific cases, darkness can be used, but not during all culture growth". Everything is exactly the opposite - plant cell cultures are heterotrophic cells, and are usually grown in the dark. Light is used only when growing mycostrophic or autotrophic cell cultures. Various variants of the regulatory action of light are possible, but these are special research tasks. At the same time, for cultures of producing cells, the need for light regulation is rather a minus than a plus - how can this be done in a bioreactor?

Response 3. The reviewer comments that cells are heterotrophic and that they are generally grown in the dark (this is not entirely correct). In many reports is well supported that plant cell cultures grow adequately under photoperiod conditions (generally 16 light hours and 8 darkness). As discussed early, it is rare to find in the literature that suspension cell cultures are grown in the dark. In fact, it was demonstrated with our experience in plant cell cultures and for some of several reports, for example:

  • Jamil et al., (2018). 3 Biotech, 8(8), 1-14. https://dx.doi.org/10.1007%2Fs13205-018-1336-6.
  • Sánchez-Ramos et al. (2020). Plants, 9(10), 1398. https://doi.org/10.3390/plants9101398.
  • Ahmad et al. (2020), Plant Cell Tissue and Organ Culture, 143, 565–571, https://doi.org/10.1007/s11240-020-01941-z
  • Sánchez-Ramos et al. (2018), Molecules 2018, 23(6), 1258; https://doi.org/10.3390/molecules23061258.
  • Arano-Varela, et al. (2020), South African Journal of Botany, 135, 41-49, https://doi.org/10.1016/j.sajb.2020.08.005.
  • Sathish, et al., (2019). Plant Biotechnology Reports, 13(6), 613-621. https://doi.org/10.1007/s11816-019-00555-y.

Many more reports can be found in the literature.

            And as mentioned in the first review, light is an essential factor in the growth of cell cultures. In fact, it is reported that LED technology is favoring the development of better efficiencies in plant tissue culture [Batista et al., 2018, In Vitro Cellular & Developmental Biology – Plant, 54(3), 195-215. https://doi.org/10.1007/s11627-018-9902-5].

Only in specific cases, darkness can be used, but not during all culture growth. In fact, it has been reported that under dark conditions, the production of some compounds is negative compared with culture grown under light conditions, for example: Behbahani et al., (2011). Scientia Agricola, 68(1), 69-76 (https://doi.org/10.1590/S0103-90162011000100011). In addition, our study was no aimed to evaluate variants of lighting conditions because it is known that light is essential for any in vitro culture. Moreover, our in vitro cultures have grown suitably under light conditions.

            Regarding bioreactors, there are generally bioreactors that already include accessories to incorporate light, but in many cases, bioreactors can be adapted in a room with controlled light. In our study, we do not use bioreactors.

Point 4: The authors explained the method of maintaining the cell culture (however, very confusingly) - "the initial inoculum size for a 1 L total volume flask is 12 g contained in 200 mL (5%)", that is, it can be understood that a one-liter flask with 200 ml of medium was introduced 10 ml of inoculum with 0.6 g of cells.

Response 4: The methodology explains that the cells were filtered to remove excess culture medium. Once filtered, the fresh weight cell biomass was transferred to flasks containing liquid culture medium. Page 16, lines 404-406. In 1-L capacity flasks containing 200 mL of culture medium, they were always inoculated with 12 g of cell biomass in fresh weight, aliquots were not used.

Point 5:  - however, in the manuscript the growing conditions are completely different "Kinetic growth was carried out in Erlenmeyer flask of 125 mL containing 25 mL of MS liquid medium and 1.5 g of cells."  Firstly, the degree of aeration of the culture decreases with a decrease in the volume of flasks and the volume of the suspension should be reduced, and secondly, what does"1.5g of cell" mean? Is this fhesh weight?

Response 5: Yes, the growth kinetics were performed in Erlenmeyer flasks of 125 mL capacity containing 25 mL of liquid MS culture medium and all were inoculated with 1.5 g of fresh weight cell biomass, as clearly explained in the manuscript. Page 16, lines 401-402.

            In several reports, the growth kinetics is reported as g of biomass in fresh weight/L of culture medium as in our study. For example:

Sánchez-Ramos et al. (2020). Plants9(10), 1398.

del Pilar Nicasio-Torres et al. (2012). Acta physiologiae plantarum34(1), 307-316.

Pérez‐Hernández, et al., 2019. Engineering in Life Sciences19(3), 196-205.

            In our laboratory, all cultures are grown at 115 rpm and we have reported studies under these conditions. Moreover, the inoculum size is not the same in all flask sizes, the inoculum size is based on the volumetric capacity of each flask and the volume of culture medium; that is, it is carried out proportionally, for example, the 125-mL flasks containing 25 mL of MS culture medium are inoculated with 1.5 g of fresh weight cell biomass, the 250-mL flasks containing 50 mL of MS are inoculated with 3 g, 500 mL flasks containing 100 mL of MS are inoculated with 6 g, and flasks with 1000 mL containing 200 mL are inoculated with 12 g in fresh weight. That is, the inoculum size is proportional to the total volume of the flask and the culture medium contained therein, therefore, the degree of aeration is not affected. Our cultures grow adequately in the different sizes of flasks, and our inoculum sizes are in accordance with several reports in the literature.

  • Jamil et al., (2018). 3 Biotech8(8), 1-14. https://dx.doi.org/10.1007%2Fs13205-018-1336-6.
  • Sánchez-Ramos et al. (2020). Plants9(10), 1398. https://doi.org/10.3390/plants9101398.
  • Ahmad et al. (2020), Plant Cell Tissue and Organ Culture, 143, 565–571, https://doi.org/10.1007/s11240-020-01941-z
  • Sánchez-Ramos et al. (2018), Molecules2018, 23(6), 1258; https://doi.org/10.3390/molecules23061258.
  • Arano-Varela, et al. (2020), South African Journal of Botany, 135, 41-49, https://doi.org/10.1016/j.sajb.2020.08.005.
  • Sathish, et al., (2019). Plant Biotechnology Reports13(6), 613-621. https://doi.org/10.1007/s11816-019-00555-y.

Point 6: Then, based on the graph, the DW FW ratio is 1: 20 (1.5 g in 25 ml, that is, in 1000 ml - 60 g, the initial density according to the graph is 3 g / l) and in the hospital the fresh weight at the time of inoculation (18 g by dry weight) should be 360 g / l, which is practically impossible.  This question was in the first review - and the authors decided not to answer it. By the way, this is a good illustration of the benefits of determining fresh weight: The opinion of the authors of "Measuring the dry weight is more suitable than the fresh weight, since there is a lot of variation and the results in fresh weight are not correct due to the variation in moisture content." is certainly true, and it necessary to measure both dry weight and fresh weigh  at least to avoid obvious mistakes.

Response 6: Yes, in the growth curve, an initial inoculum of 3 g/L is mentioned, but to make the growth curve the dry weight data were considered because dry weight is more appropriate than fresh weight. In fact, as mentioned above, several studies support ours, and we have published several studies in specialized journals, such as Plants, Plant Cell Tissue and Organ Culture, Industrial Crops and Products, South Africa Journal of Botany, Molecules, e.g. Sánchez-Ramos et al. (2020). Plants9(10), 1398; Bernabé-Antonio, et al. (2015). Plant Cell, Tissue and Organ Culture (PCTOC)120(1), 221-228.; Cisneros-Torres et al. (2020). Revista Mexicana de Ingeniería Química19(1), 59-70.

Point 5: An accurate calculation of growth parameters is certainly useful, but if this is done for only one growth curve, it makes little sense since the curve is shown. These characteristics - and most importantly, their spread - are important for assessing the stability of crop growth when analyzing at least 3 - 4 curves.

Response 5: Yes, in the methodology section it is mentioned that the cell cultures were frequently subcultured and maintained for at least 6 months. Page 16, lines 406-407.             During this time, no visual changes were observed in our cell culture, therefore, we ensure that our cultures are totally stable; in fact, to date we still have them under maintenance for current projects, and no changes have been observed.

            Regarding to the growth curve, ¡Yes!, a growth curve was plotted, but 3 experiments were performed at different times and averages were obtained (these are mentioned on page 19, line 514), so the calculated parameters are correct. However, our main objective was not only to characterize the culture growth. Furthermore, our already reported works as well as those of other authors (mentioned above) support this study.

Point 6: The authors believe that “GC – MS is a classical technique used in natural product research to separate and identify volatile compounds (e. G. Essential Oils). Thus, in the case of extracts with non-volatile compounds, the alkylsilyl derivatives have found extensive use for imparting high volatility and stability to the compounds for GC – MS analysis ”As for essential oils, the authors are absolutely right, but about the rest (despite the fact that The reviewer respects GC-MS very much and worked for a long time on the first GC-MS LKB GC-MS 2091 device ..) unfortunately not - at least from the molecular weight limit, not to mention possible artifacts when obtaining derivatives .. It is obvious that now HPLC-MS is the classic method of such work. For GC – MS analysis, there are clear boundaries of application where it is adequate.

Response 6: Yes, GC–MS is a classical technique used in natural product research to separate and identify volatile compounds. Our methodology used is supported by previously reported studies. In fact, there are specialized journals in Gas Chromatography (Journal of Chromatographic Science, Journal of Chromatography A and Journal of Chromatography B-Analytical Technologies in the Biomedical and Life Science) with a high impact factor, in which many studies on natural products have been published.

            On the other hand, Essential Oils are a type of compounds analyzed by GC-MS but they are not the only ones, also other type of volatile compounds have been analyzed using this technique, e.g. quinolines, terpenes, coumarins, pyranocoumarins and others [Marrelli et al. (2021) Plants, 10(1), 123; Naama-Amar et al. (2020). Plants, 9(1), 72; Nazlic et al. (2020). Plants, 9 (12), 1646; Stasiak et al. (2020). Plants, 9 (12), 1675; Gad et al. (2021). Plants, 10 (1), 124; Many reports can be found in the Plants journal or others].

            In our estudy, the main idea was determining the phytochemical profile of the organic extracts by spectrophotometric and gas chromatography-mass spectrometry (GC-MS) methods, in GC-MS method the extracts were derivatized with BSTFA to identify as many compounds as possible, according to the previously reported methodology that was cleared in the first review [Miranda et al. (2017). PLOS ONE, 12(6), e0179268; Alonso-Castro et al. (2018). Journal of Ethnopharmacology, 224, 314; Harvey and Vouros (2019). Mass Spectrometry Reviews, 00, 1]. As mentioned, GC-MS is a classic technique used in natural product research to separate and identify volatile compounds, but it is not the only one, HPLC-MS is also an important technique used in natural product research, however, GC-MS is still a widely used and accepted technique to separate and identify volatile compounds, as we have shown with the references mentioned to natural products and alkylsilyl derivatives.

Point 7: It makes no sense to discuss the authors' comments on the methodology and principles of the chemical part of the work, the main points were said in the first review. The only strange logic is “About of the phytosterols, we included them because their antifungal activity has been reported, eg, stigmasterol and sitosterol have also been reported to be effective against phytopathogenic fungi” .. That is, the choice of analytes is based on their antimicrobial activity and about the rest of the analyzed compounds have data?

Response 7: We are pleased to report that all comments on the methodology and principles of the chemical part of the work were answered point by point in the first review. As we mentioned in the first review, phytosterols were included because their antifungal activity has been reported; for example, stigmasterol and sitosterol have also been found to be effective against phytopathogenic fungi. In the present investigation, we determined the phytochemical profile of the organic extracts using spectrophotometric methods and GC-MS. Also, in the discussion section, we mention the bioactivity of the major compounds: Page 10, Lines 247-251 and page 15, Lines 339-362. On the other hand, the addition of the bioactivity of all the identified compounds would not be suitable for a regular research article, but rather for a review article.

Point 8: In conclusion, the analysis was carried out for only one crop cultivation and only once. Are there any guarantees that in another cultivation cycle the composition and amount of the analyzed substances will not be different - and, accordingly, their biological activity? Similarly - is there any guarantee that the main antimicrobial activity is not provided by those compounds that cannot be determined by GC MS?

Response 8: In the Statistical Analysis part, we mentioned that all the experiments were conducted in triplicate and this was cleared (page 19, line 514). Of course, is possible that in some cases the main antifungal activity is not provided by the compounds identified by GC MS, for this reason we considered important to determine the phytochemical profile by spectrophotometric and GC-MS methods. In addition, one of the great advantages of plant cell culture is that they are kept cultivated under controlled conditions of light and temperature throughout the time, so it is considered that the variations may be minimal. [Mulabagal, V., Tsay, H. S. 2004. Int J Appl Sci Eng2(1), 29-48.]. In conclusion, we greatly appreciate all the valuable recommendations of the reviewer to improve the manuscript. We have the pleasure to inform you that we were able to include all your observations in the improved manuscript.

Point 9. Unfortunately, the authors made only minor changes to the text of the manuscript, and publication in Plants does not recommended.

Response 9. We have responded and acted upon all suggestions/comments made by reviewers and showed evidence to support our work.

Reviewer 2 Report

Dear authors,

the article you submitted to me proved to be interesting and engaging. The idea of ​​conserving the wild plant population is particularly commendable. Biotechnological solutions are particularly important in order to use natural resources as efficiently as possible and thus minimize damage to nature. You have shown in your study how to effectively extract the active components of plants and at the same time check their antifungal properties. despite the interesting content, several errors occurred in the work.

In Figure 3 please correct the name of the right axle. You mention TFL everywhere in the text, and the axis is called FLT. I understand this is just a typing mistake.

In line 199 in the sentence The methanolic extract of ... it is difficult to understand what that effective solvent is. It should be like methanol, but the sentence is confusing.

Line 482 should contain an agar-diffusion methot, a letter is missing.

In section 2.5 it is difficult to understand which extracts (hexane, dichloromethane or methanol) were used to determine antifungal activity. Only after reading the methodological part does it become clearer.

I also have a couple of questions about chromatographic analyzes:

  1. How did you evaluate the components of non-volatile methanolic extracts? Maybe you did some procedure, without derivatization, to get rid of them. Methanol is known to be an excellent solvent for non-volatile components as well, and the latter can contaminate the column.
  2. 2. How could you explain the presence of phthalates in fatty and defatted hexane and some dichloromethane extracts?

Good luck

Author Response

Response to Reviewer 2 Comments

The article you submitted to me proved to be interesting and engaging. The idea of ​​conserving the wild plant population is particularly commendable. Biotechnological solutions are particularly important in order to use natural resources as efficiently as possible and thus minimize damage to nature. You have shown in your study how to effectively extract the active components of plants and at the same time check their antifungal properties. despite the interesting content, several errors occurred in the work.

Point 1: In Figure 3 please correct the name of the right axle. You mention TFL everywhere in the text, and the axis is called FLT. I understand this is just a typing mistake.

Response 1: This was done. The word “FLT” was substituted by “TFL” in Figure 3 and on Page 5, line 170.

Point 2: In line 199 in the sentence The methanolic extract of ... it is difficult to understand what that effective solvent is. It should be like methanol, but the sentence is confusing.

Response 2: The sentence was rewritten as follows: “Other studies carried out on branches of Severinia buxifolia also obtained a higher extract yield (33.2%) using methanol”.  Page 6, lines 198-199.

Point 3: Line 482 should contain an agar-diffusion methot, a letter is missing.

Response 3: The word “gar” was corrected as “agar”. Page 18, line 496.

Point 4: In section 2.5 it is difficult to understand which extracts (hexane, dichloromethane or methanol) were used to determine antifungal activity. Only after reading the methodological part does it become clearer.

Response 4: The paragraph was corrected. The beginning of the sentence of Section 2.5 “As an assay of biological activity, we evaluate the antifungal potential of the extracts on phytopathogenic fungi available in our laboratory (Rhizoctonia solani and Sclerotium cepivorum).” was rewritten for clarity as fallows: “As an assay of biological activity, we evaluate the antifungal potential of hexanic (fatty and defatted), dichloromethane and methanolic extracts from sapwood, heartwood, leaves, and suspension cell cultures on phytopathogenic fungi available in our laboratory (Rhizoctonia solani and Sclerotium cepivorum).” Page 13, lines 293-296.

Point 5: I also have a couple of questions about chromatographic analyzes: 1.How did you evaluate the components of non-volatile methanolic extracts? Maybe you did some procedure, without derivatization, to get rid of them. Methanol is known to be an excellent solvent for non-volatile components as well, and the latter can contaminate the column.

Response 5: Yes, methanol is an excellent solvent to obtain non-volatile components, e. g. carbohydrates, phenols and flavonoids. All the crude extracts were derivatized with BSTFA to identify as many compounds as possible by GC-MS analysis. In the specific case of methanol extracts, carbohydrates and polyols were identified as alkylsilyl derivatives by GC-MS as showed in the Table 4 of the manuscript (Pages 11 and 12). However, in the chromatogram of methanol extracts (depending on the source of the extract) approximately 4-57% of the peaks were not possible to identify and we complement the phytochemical characterization of the extracts through the spectrophotometric quantification of phenolics total and flavonoids content. In future studies, we will be reporting other works to isolate and characterize by NMR the majority compounds of the most active extracts.

Point 6: 2. How could you explain the presence of phthalates in fatty and defatted hexane and some dichloromethane extracts?

Response 6: In our work, laboratory glassware was used in all experiments and in the extraction process high purity solvents (ACS reagent) were used (checked before use), for this reason we are sure that the phthalates are not a contamination. Some studies report that the phthalates are present inclusive in the cell culture extracts (CSC) and we never used plastic material in the biotechnological and chemical experiments. For instance, some authors have been reported phthalates in plants species [Ruikar et al. (2011). Chemistry of Natural Compounds, 46(6), 955; Shobi et al. (2018). Journal of Applied Biotechnology & Bioengineering, 5(2), 97] for this reason we included them in the results.

Reviewer 3 Report

The manuscript is recommended for publication. Several changes are recommended, and some clarifications are required.

Page 2. Line 62 Sclorotium correct in Sclerotium.

Page 2 Line 53, page 3 Line 82, page 6 Line 196, page 6 Line 208, page 14 Line 280, page 16 Line 346, page 23 Line 700 Fabaceae, should be typed in italics. According to rules italicize family, genus, species, and variety or subspecies.

Page 4 Line 122-123, 127, mg/L, or mg L? Please, clarify.

Page 10 Line 247 Robinia pseudoacacia, please italicize.

Page 16 Line 346 Dahlstedt glaziovii? Please, clarify.

Page 19 Line 482...... the gar disk-diffusion method? Did you mean the agar disk-diffusion method?

Page 21 Line 553, page 23 Line 647, page 23 Line 657, page 23 Line 673 In vitro, please italicize.

Page 22 Line 646 Fabaceae-caesalpinioideae, please italicize.

Page 23 Line 704 In vitro and in vivo, please italicize.

Author Response

Response to Reviewer 3 Comments

The manuscript is recommended for publication. Several changes are recommended, and some clarifications are required.

Point 1: Page 2. Line 62 Sclorotium correct in Sclerotium.

Response 1: This was done. The name “Sclorotium” was changed by “Sclerotium”. Page 2, line 71.

Point 2: Page 2 Line 53, page 3 Line 82, page 6 Line 196, page 6 Line 208, page 14 Line 280, page 16 Line 346, page 23 Line 700 Fabaceae, should be typed in italics. According to rules italicize family, genus, species, and variety or subspecies.

Response 2: We reviewed the rules of the International Code of Nomenclature for algae, fungi, and plants 2018 (https://www.iapt-taxon.org/nomen/main.php), and The Plant List database (http://www.theplantlist.org/) and the family names was corrected. Family “Fabaceae” was changed by italicized name “Fabaceae”. Page 1, line 17; page 2, line 52; page 3, line 83; page 6, lines 196 and 207; page 10, line 251; page 13, line 289; page 15, line 358.

Point 3: Page 4 Line 122-123, 127, mg/L, or mg L? Please, clarify.

Response 3: This was done. The correct unit is mg/L. The entire text was revised and "mg L" was substituted by "mg/L". Page 4, lines 123, 124 and 128.

Point 4: Page 10 Line 247 Robinia pseudoacacia, please italicize.

Response 4: This was done. The name “Robinia pseudoacacia” was italicized “Robinia pseudoacacia”. Page 10, line 251.

Point 5: Page 16 Line 346 Dahlstedt glaziovii? Please, clarify.

Response 5: It was a typo. “Dahlstedt glaziovii” was corrected as “Dahlstedtia glaziovii”. Page 15, line 358.

Point 6: Page 19 Line 482...... the gar disk-diffusion method? Did you mean the agar disk-diffusion method?

Response 6: Yes, it was corrected as “agar disk-diffusion method”, Page 18, line 496.

Point 7: Page 21 Line 553, page 23 Line 647, page 23 Line 657, page 23 Line 673 In vitro, please italicize.

Response 7: This was done. All text was reviewed and corrected where necessary.  The words “In vitro” or “in vitro” was modified in italic format “In vitro or in vitro”. Page 20, line 567; page 22, lines 671, 687 and 718.

Point 8: Page 22 Line 646 Fabaceae-caesalpinioideae, please italicize.

Response 8: This was done. The name “Fabaceae-caesalpinioideae” was italicized asFabaceae-caesalpinioideae. Page 21, line 660.

Point 9: Page 23 Line 704 In vitro and in vivo, please italicize.

Response 9: This was corrected above. Page 22, line 718.

Reviewer 4 Report

In the presented work, suspension cultures of Eysenhardtia platycarpa have been established and evaluated for chemical composition and antifungal activity. The described cell culture experiments are a basic ones and included callus initiation, followed by establishing cell suspension and determining its growth kinetics. The obtained culture was examined for total phenolic and total flavonoid contents which are also considered basic parameters characterizing plants’ chemistry. Additionally, different types of extracts from cell suspension and intact plant material were prepared and examined by examined by GC-MS, which gives some insight into chemical composition of the plant. The manuscript also contains some elements of novelty, since the genus Eysenhardtia has not been extensively studied so far. In my opinion, the paper requires minor revision before it can be published in plants. Apart from specific points listed below, the paper requires minor language editing.

Specific points:

lines 37-52: in my opinion, this part can be substantially shortened. The presented information are well known and there is no need to e.g. provide a list of compounds produced using in vitro techniques.

lines 47-47: this may be true for some compounds but it is not a general rule

lines 53-64: please provide a sound rationale for the conducted experiments. Is the species endangered or extensively exploited so that in vitro techniques are necessary to exploit its potential?

lines 117-119: again, this is a well known observation. Such comments can be omitted, in my opinion

Author Response

Response to Reviewer 4 Comments

In the presented work, suspension cultures of Eysenhardtia platycarpa have been established and evaluated for chemical composition and antifungal activity. The described cell culture experiments are a basic ones and included callus initiation, followed by establishing cell suspension and determining its growth kinetics. The obtained culture was examined for total phenolic and total flavonoid contents which are also considered basic parameters characterizing plants’ chemistry. Additionally, different types of extracts from cell suspension and intact plant material were prepared and examined by examined by GC-MS, which gives some insight into chemical composition of the plant. The manuscript also contains some elements of novelty, since the genus Eysenhardtia has not been extensively studied so far. In my opinion, the paper requires minor revision before it can be published in plants. Apart from specific points listed below, the paper requires minor language editing.

Specific points:

Point 1: lines 37-52: in my opinion, this part can be substantially shortened. The presented information are well known and there is no need to e.g. provide a list of compounds produced using in vitro techniques.

Response 1: We review the indicated section and agree to shorten it and the sentence “In fact, some plants extract or compounds such as ajmalicine, anthraquinones, berberine, caffeic acid, ginsenoside, nicotine, rosmarinic acid, shikonin have been produced using the plant cell, tissues, and organ culture [5].” was deleted.

Point 2: lines 47-47: this may be true for some compounds but it is not a general rule

Response 2: The paragraph “Although plants can be cultivated in the field, many of them need several years to be harvested, in addition, the yields of bioactive compounds and biological activity of farmed or wild plants are low when compared with cultured plant cell” was slightly rewritten as follows “Although plants can be cultivated in the field, many of them need several years to be harvested, and in many cases, the yields of bioactive compounds and biological activity of cultivated or wild plants are lower compared with cultured plant cells”. Page 2, lines 45-48.

Point 3: lines 53-64: please provide a sound rationale for the conducted experiments. Is the species endangered or extensively exploited so that in vitro techniques are necessary to exploit its potential?

Response 3: The paragraph “In this regard, Eysenhardtia platycarpa (Fabaceae), is a plant with a wide variety of uses such as manufacture of utensils, furniture, as well in traditional Mexican medicine” was slightly modified as follows “In this regard, Eysenhardtia platycarpa (Fabaceae) is a wild plant extensively exploited as firewood, fodder, or to manufacture utensils and furniture such as “equipales” and fences; in addition, in traditional Mexican medicine, an infusion prepared from the wood is used against kidney and gallbladder diseases [7,8,9]. All these uses, as well as forest fires, are causing a decline in wild populations.”. Page 2, lines 52-56.

Point 4: lines 117-119: again, this is a well known observation. Such comments can be omitted, in my opinion

Response 4:  We reviewed the paragraph (lines 117-119) and agreed that it is necessary because the articles are for a wide variety of audiences and readers. The comment was no omitted. Page 4, lines 118-120.

This manuscript is a resubmission of an earlier submission. The following is a list of the peer review reports and author responses from that submission.

Round 1

Reviewer 1 Report

The manuscript deals with the in vitro cultures of cell suspension cultures of E. platycarpa and screening of valuable metabolites in its extracts, as well as antifungal activity. The Authors applied phytochemical approach to study the profile of metabolites. Conclusions adequate to the conducted research. In my opinion the manuscript is novel, the authors obtained new unique experimental data. The idea of research was very good.

Nevertheless some issues must be addressed before publication:

  1. page 3, line 81 - in Table 1 symbols a and b are not introduced (i.e. in Callus induction results) and they are not easy to interpret; 
  2. p. 4, l. 122 - substrate concentration dropped down to 0 after 12 days of culture, but almost no effect on biomass concentration maybe seen up to 18th day of the process - such effects should be explained; 
  3. p. 13, l. 249 - error bars are missed for Cercobin (a) results;
  4. list of symbols and abbrieviations should be prepared and incorporated into the manuscript; 
  5. some grammar errors require corrections. 

Reviewer 2 Report

the paper has been already published in other journal

preprints.org > life sciences > biochemistry > doi: 10.20944/preprints202011.0167.v1

Reviewer 3 Report

Comments to the manuscript entitled ‘Establishment of a Cell Suspension Culture of Eysenhardtia platycarpa: Phytochemical Screening of extracts and Evaluation of Antifungal Activity’ (Manuscript number: plants-1006066) written by Antonio Bernabé-Antonio, Alejandro Sánchez-Sánchez, Antonio Romero-Estrada, Juan Carlos Meza-Contreras, José Antonio Silva-Guzmán, Francisco Javier Fuentes-Talavera, Israel Hurtado-Díaz, Laura Álvarez  and Francisco Cruz-Sosa.

In the reviewed manuscript authors present study on establishment of cell suspension cultures of medicinal plant, together with analyses of chemical composition of extracts obtained from cell suspension and also from different plant parts. Moreover, authors tested also antifungal activity of obtained extracts against plant fungi. Although presented study seems very interesting, authors did not avoid some omissions and errors in the manuscript, so I recommend manuscript to publication but after major improvement.

General comments:

English language of manuscript should be checked by native speaker.

Manuscript lacks of scientific hypothesis.

Discussion should be more profound, especially chapter 2.3 and 2.4, e.g. try to explain why some solvents were better extractants.

Conclusions must be rewritten. In present form this chapter is summarizing the results not conclusions.

There is lack of some statistical analyses that must be complemented.     

Particular comments:

Abstract

Line 23 – authors write about sapwood and heartwood extracts composition without mentioning earlier that intact plant was also analysed.

Line 28 – ‘higher for Slerotium cepivorum...’ higher than what ???

Introduction

            Line 43 – ‘Although some plants can be farmed...’ ??? – this is oversimplification

Results and discussion

            Line 69 – ‘Seeds had 98% germination...’ ? – language style

Line 162 – there is information about TPH content in leaves in the result chapter on in vitro cultures. Is should be clearly stated that you compare phenols content in in vitro cultures and leaves.  

Line 171 – ‘...conventional culture.’ What do you mean??

Line 173 – ‘...the highest dry weight yields’ There is no information in material and methods how you calculated this dry weight.

Table 2 – there is no statistical analyses of these results so how can you tell what was higher or lower?

Table 3 and 4, Figures 4 and 5 – lack of statistical analyses, the same as in Table 2, how can you interpret the results? It must be done.

Figure 6 and 7 – x axis description ‘Plant extract resource’ is not appropriate considering cercobin and ethanol.

Material and Methods

Line 422 – How you calculated percentage inhibition of mycelial growth? Description should be added.

Statistical Analyses – why did you not statistically analyse abundance percentage of chemical constituents and yield of extracts? It must be done to properly interpret the results.

Reviewer 4 Report

The manuscript presents the results of work on obtaining a suspension culture of Eysenhardtia platycarpa cells, a valuable medicinal plant in Mexico, as well as its growth and biosynthetic characteristics. The authors have done a fairly large amount of work and received a significant number of results, however, the publication of the manuscript in the presented form in the journal Plants is impossible.

  1. Section on the obtain and characteristics of cell culture.

Nowadays, obtaining a plant cell culture is quite standard, and rather technical than scientific work. Since the task was to obtain and characterize a suspension cell culture, a detailed description of the routine procedure for obtaining calli, in our opinion, is unnecessary and of little interest. In addition, there are obvious errors in this description. For example in table 1 in the absence of 2,4D and KIN Callus induction in the environment is 50%, and in the absence of NAA and KIN - 75%  (that is, in any case - on the same hormone-free environment) - but the maximum variation of the obtained  results (according Tabl. 1) is 8.8 %

 At the same time, the obtained suspension cell cultures are clearly insufficiently characterized and with serious errors.

First, the very initiation of suspension cell cultures. It is unclear why the lighting was used. This is rather a negative factor of cultivation, and if this was the subject of research, then it is necessary to use the dark variant as a control.

Second, the characteristics of the resulting cell culture. Only one graph of the growth curve for dry biomass and the dynamics of absorption of sucrose from the medium are shown. The regularities of the absorption of sucrose by cells have been studied very well for a large number of cultures and, in the absence of special tasks, are of no interest or novelty.

At the same time, the parameters of growth according to different criteria (fresh weight, number of cells) are not presented in the manuscript, but it is very important. Suffice it to say that in the Methodology section (line 352) it is said that 25 ml. The medium was supplemented with 1.5 g of cells (obviously, fresh weight), that is, the initial concentration according to this criterion was 60 g / l. From graph 2 it follows that the initial concentration of cells in dry biomass is about 3 g / l. It follows from this that the hydration of the cells is about 95%, which indicates that the culture is not in the best state (in any case, with such a ratio of dry and wet cell biomass, a cytological analysis of cells is desirable). It can be easily calculated that at the time of inoculation (on the 14th day - 18 g by dry weight, according to Fig. 2), the concentration of fresh biomass was 360 g / l, which is impossible.

Further, the authors give in the text of the article the growth parameters of the obtained cell culture - the duration of the lag phase, the specific growth rate, the economic coefficient, but they do not give how they calculated it. Checking the presented results using standard methods (for example, using a semi-logarithmic scale for the data shown in Fig. 2) indicates serious errors in the determination and calculations - according to this growth curve, the lag phase is 1.5 days, not 4 days, the specific growth rate is 0.27 day-1, not 0.21 (and the authors forgot to indicate its dimension); accordingly, the doubling time (td) was not 3.36 days, but 2.55 days. And finally, the Economic coefficient Y (in the author's version - "yield") is not 0.621 g dry biomass / g sucrose, but 0.5 (the authors forgot to subtract the biomass of the inoculum).

The most important thing is that even a minimal optimization of parameters cultivation was not carried out in the work — the initial cell density (3 g by dry biomass) is clearly overestimated. The flask volume / medium volume ratio used 5: 1, which is clearly insufficient (the optimal variant is usually 10: 1), and most likely there was oxygen limitation for cells . The culture was subcultured too late (as follows from the text, every two weeks), - however, on the 14th day, the culture was in the stationary growth phase for at least 4 days, while replanting must be carried out at the end of the exponential growth phase - the beginning of the slowdown phase, that is, in this case - on 9-10 days of cultivation.

All of the above suggests that the resulting cell culture is practically not characterized, its cultivation is not optimized, and therefore, with a high probability, the culture is not stable and its use for reliable phytochemical analysis is not correct. not correct.

  1. Section on phytochemical analysis. The main idea of ​​the analysis is not entirely clear (analysis of everything that can be determined by the GC-MS method?) - and the almost complete absence of analysis of the results obtained. Metabolomics using GC-MS is an attractive idea, but not feasible due to too many limitations of this method.

This section raises no less questions than the cultural part of this work

  • a) A strange choice of extraction methods, without any assessment of its completeness. Silanization of dry extracts, even if we assume that all components have turned into volatile silanes, cannot be considered a successful approach, since one cannot be sure that all analytes in the evaporation chamber of the chromatograph enter the column and do not settle on the walls of the liner.
  • b) What the authors call "semi-quantitative data", unfortunately, cannot be such, since they represent only mass% of the sum (all peaks? or only identified?), while no internal standards (each group of compounds needs its own) is used, nor the gravimetry of the dry extract, which at least approximately made it possible to estimate the amount of substances under discussion.
  • c) the method chosen by the authors is not suitable even as a primary screening, since the detected metabolites - free FAs, wax esters, monosaccharides - indicate that no complete extraction took place, and the authors studied only "unbound" metabolites or degradation products during sample preparation lipids, carotenoids, etc.

Probably, phytochemical analysis should still be carried out according to generally accepted methods and the logic of such studies.

- Select the pools of metabolites that the authors want to investigate and optimize an efficient extraction system for them. As a rule, two-phase systems of J. Folch, or E. C. Bligh and W. J. Dyer are used for these purposes: However, the volatile organic compounds, which the authors list, require special extraction systems and sample preparation methods.

    - Using preparative TLC or solid-phase extraction on silica (better, on C-18 cartridges), fractionate the obtained extracts (after selecting the optimal method) by classes: for example, neutral lipids (sterol esters, hydrocarbons, wax esters, glycerolipids), polar lipids , carotenoids, etc.

 - For each of the obtained fractions, carry out adequate sample preparation using internal standards. For example, sterols in cells can be in free form, in glycoside form, or in ester form. Free sterols can be analyzed free or silanized; for sterol glycosides, hydrolysis is required, for esters, saponification, and each fraction must be analyzed separately. For polar lipids, division into classes is necessary: ​​at least glycolipids, sulfolipids, phospholipids, followed by analysis of each of them.

The methods not optimized by the authors and the lack of analysis of the results obtained led to many erroneous and incorrect results. For example, the phthalates and other plasticizers discovered by the authors are obviously artifacts of the analysis, and they should not have been included in the summary tables!

The FA composition of the extracts raises many questions - for example, there is surprisingly little 18: 3 FA in the leaves, and the main one is palmitic. Photosynthesis with such a ratio of LC is impossible! Noticeable amounts of azelaic acid also indicate the processes of oxidation and degradation of both FA and lipids containing them.

The phytosterols discovered by the authors - sitosterol and stigmasterol - are primary metabolites, components of plant cell membranes - and, therefore, are present in all plants. Did it make sense to include them in the pivot table of detected compounds? Why are they absent in their leaves (isn't it because they can be there in the form of glycosides)? Why are phytosterols found in at least two fractions (lack of control over the completeness of extraction?) Finally, why is the third main phytosterol, - campesterol, - not found?

And the most important thing:  extraction from raw materials preliminarily dried in air at an elevated temperature is not suitable for all metabolites. In this case, the analysis of the content of volatile components gives an idea only of the miraculously not evaporated residual metabolites; Such drying gives distorted results on the composition of FA and lipids, carotenoids, wax esters, and sterols. For metabolomic tasks, extraction must be carried out either from fresh raw materials (and very quickly!) -  or preserved by one method or another.

All of the above does not allow recommending the manuscript in Plants.